## Registered report

psychology/behaviour

diversity, herding, decision-making, group behaviour, social interaction

**Author for correspondence:**
Joaquin Navajas
e-mail: joaquin.navajas@utdt.edu

# Diversity of opinions promotes herding in uncertain crowds

Joaquin Navajas[1,2,3], Oriane Armand[4,5], Rani Moran[7,8], Bahador Bahrami[5,6,9] and Ophelia Deroy[4,5,10]

[1]Laboratorio de Neurociencia, and [2]Escuela de Negocios, Universidad Torcuato Di Tella, Buenos Aires, Argentina
[3]National Scientific and Technical Research Council (CONICET), Buenos Aires, Argentina
[4]Faculty of Philosophy, [5]Munich Centre for Neuroscience, and [6]Faculty of Psychology and Educational Sciences, Ludwig-Maximilians-Universität München, Munich, Germany
[7]Max Planck UCL Centre for Computational Psychiatry and Ageing Research, London, UK
[8]Wellcome Centre for Human Neuroimaging, University College London, London, UK
[9]Centre for Adaptive Rationality, Max Planck Institute for Human Development, Berlin, Germany
[10]Institute of Philosophy, School of Advanced Study, University of London, London, UK

 JN, 0000-0001-8765-037X; RM, 0000-0002-7641-2402;
OD, 0000-0001-9431-3136

Classic and recent studies demonstrate how we fall for the 'tyranny of the majority' and conform to the dominant trend when uncertain. However, in many social interactions outside of the laboratory, there is rarely a clearly identified majority and discerning who to follow might be challenging. Here, we asked whether in such conditions herding behaviour depends on a key statistical property of social information: the variance of opinions in a group. We selected a task domain where opinions are widely variable and asked participants ($N = 650$) to privately estimate the price of eight anonymous paintings. Then, in groups of five, they discussed and agreed on a shared estimate for four paintings. Finally, they provided revised individual estimates for all paintings. As predicted (https://osf.io/s89w4), we observed that group members converged to each other and boosted their confidence following social interaction. We also found evidence supporting the hypothesis that the more diverse groups show greater convergence, suggesting that the variance of opinions promotes herding in uncertain crowds. Overall, these findings empirically examine how, in the absence of a clear majority, the distribution of opinions relates to subjective feelings of confidence and herding behaviour.

## 1. Introduction

People tend to believe what others believe and do what others do. This uncoordinated convergence of thoughts or behaviours in a

group, known as 'herding' [1,2], finds its most famous and dramatic evidence in the experiments conducted by social psychologists in the aftermath of the Second World War. An extensive literature has demonstrated how people might go against their own private evidence and conform to an ostensibly wrong but unanimous mob [3–6]. Numerous studies have examined the many contextual factors that may affect the costs and benefits of herding for social individuals when making uncertain decisions [7–9].

What has been far less investigated is an equally urgent and practical aspect of uncertainty that is specific to social interaction: in many situations, for example when people discuss prices or quantities, beliefs may lie in a continuum and it might be hard or even impossible to define the majority opinion. So, in these situations, interacting with a heterogeneous crowd requires that we actually infer the statistical properties of the distribution of opinions before we can decide whether to conform to it or not. In classic studies, people responded to unanimity [10,11]: there was no uncertainty in the crowd's position. Similarly, in more recent studies, people are given a summary statistic like the mean of everyone else's opinion [9], or their opinions are offered in a numerical representation [12]. But whether, and to what extent, humans infer and react to statistical properties of social information in group settings is still unknown.

Here, we examine how opinion and uncertainty might change as a result of interacting with a group of uncertain individuals. We ask if we might detect herding effects across people who, being equally uninformed as one another, simply voice their opinions. Under such a conspicuous state of uncertainty at the group level, and where herding can result from a mixture of both informational and normative elements [2,5], previous works do not even tell us if people would prefer to stick to their own views, or if they would still converge to one another.

We hypothesize that herding behaviour might depend on the variance of group opinions. Existing theories deliver conflicting predictions, which remain to be tested, about how the variance of opinions could influence herding behaviour. Several theoretical arguments suggest that herding should be minimal in groups holding more diverse beliefs. Theories such as the Social Impact Model [13] or the Social Influence Model [14] both relate the strength of social influence to the number and uniformity of influence sources. The hypothesis underlying these quantitative models is that people should be more influenced by those who are more similar to them. Social Comparison Theory [15] also predicts that the strength of social influence increases with the closeness between social sources, implying that highly similar groups could yield more convergence of opinions, while dissimilar groups would result in no change or even increased divergence [16]. The untested prediction supported by these frameworks is that groups with more variance in their distribution of opinions should be less prone to herding.

However, there is a rational argument to support the opposite prediction, that is, that more diverse groups would breed more herding. When a group of individuals hold similar opinions on a certain matter, this could indicate that beliefs are based on a common source of evidence, rendering private and social signals actually redundant. Consequently, individuals in low-variance groups have less reason to think they will benefit from following others and may continue to rely on their individual beliefs, remaining as independent as possible. Conversely, diversity of opinions could signal that information is independent across individuals, giving people a strong reason to herd. This observation has inspired qualitative [17,18] and also quantitative [19,20] theoretical formulations in fields as dissimilar as political science [17,18], computational biology [20] and microeconomics [19]. However, experimental evidence that human behaviour actually follows this pattern remains lacking. To fill this empirical gap, here we test the hypothesis that diverse groups should herd more than homogeneous groups.

Previous research also makes untested predictions about the effects of diversity of opinions on confidence. Copying the majority, when there is a clear one, can reduce one's uncertainty [21] and finding that others agree with us increases our confidence [22]. The brain's reward processing system responds positively to social agreement [23,24], and knowing that others agree with us reduces our sense of responsibility and regret for mistakes in value-based decisions [25]. These studies support the view that interaction with a more homogeneous group should boost individual confidence more than interaction with a highly diverse one.

Previous studies have focused on whether social influence increases the accuracy of crowd estimates [9,26]. Here, rather than examining the extensively studied 'crowd wisdom' effect, we focus on a different phenomenon: the relationship between herding behaviour and the initial diversity of opinions. We tested five hypotheses about how groups change their herding behaviour and confidence following a short discussion. To induce diversity and promote social influence, we used a task where (i) individuals were highly uncertain about their private opinions, while still able (and willing) to venture one, and

(ii) where they were also convinced that the problem had a correct answer. This latter concern avoided framing the task as a discussion of tastes where all subjective opinions might be as good as one another. To accomplish this issue, we asked participants to estimate the actual market price of a set of paintings previously unknown to them, and to report their confidence in those estimates. After providing individual answers, participants discussed their opinions in groups of five for a subset of paintings, with the instruction to try and agree on a consensual estimate. Finally, they revised their private estimates and confidence for all paintings. We tested our hypotheses by comparing the initial and revised estimates and confidences for the discussed and undiscussed items.

# 2. Method

## 2.1. Stimuli

We downloaded images of paintings from the webpage of a German auction site (http://www.kettererkunst.com/). We selected eight paintings from different styles and periods (e.g. abstract art, impressionism, realism etc.). All selected paintings have similar dimensions (between 30 cm and 50 cm on each side) but a wide range of market prices (from 250 EUR to 15 000 EUR). At the end of the experiment, all participants verbally confirmed whether they had observed any of these paintings before (see Data exclusion criteria).

## 2.2. Participants

Data collection took place in virtual meetings using Zoom, a widely used video-telephony software (https://zoom.us/). All participants were undergraduate or graduate students at Universidad Torcuato Di Tella (Buenos Aires, Argentina) and over 18 years old. We recruited participants via advertising at the participants list of the university. Participants with significant academic training in arts or experience in the art business were not invited to participate in this experiment. Following a power analysis based on a pilot experiment (see below), we collected data from $N = 650$ participants (i.e. 130 groups of five people). Ten research assistants, who were naive to the hypotheses of this study, assisted us with data collection across 13 sessions (each one involving 50 participants).

Participants were incentivized with a show-up fee of 200 ARS, which was the standard rate for short experiments (less than 30 min long) at Universidad Torcuato Di Tella, at the time when we performed the study. There were no further economic incentives to participate in this study. All protocols were approved by a local ethics committee (CEMIC, Centro de Educación Médica e Investigaciones Clínicas Norberto Quirno, Buenos Aires, Argentina)—Protocol 435, v. 5.

## 2.3. Procedure

Participants were asked to join the virtual meeting using a laptop or desktop computer and to always have their cameras turned on. One research assistant displayed a presentation with the images by using the 'Share Screen' tool in Zoom. Given that the study involved verbal communication between participants, we explicitly asked them to connect from a quiet room. We also requested participants to have a blank sheet of paper at hand and a pen or pencil to write down their answers.

The experimental procedure was identical in structure to a previous experiment into the effect of deliberation on the wisdom of crowds [26] and similar to a previous study about the factors enabling consensus in polarized moral issues [27]. Before the first stage, participants were provided with verbal and written instructions. Participants were sent a link to a form where they read the instructions and provided their informed consent to participate. This was implemented through Google Forms (https://www.google.com/forms/).

In the first stage of the study (figure 1a), participants privately estimated the actual market price of eight paintings. In this stage, participants had their microphones turned off. Prices were estimated in USD which is the currency used to trade imported art in Argentina. Participants were explicitly instructed to estimate the actual price, regardless of whether they liked each painting or not. This avoided framing the task as a discussion of personal willingness to pay or tastes. They also reported how confident they felt about their estimate in a Likert scale from 0 to 10. We presented the paintings one after the other for a duration of 20 s each. For each painting, participants wrote down the painting ID, their individual estimate, and their confidence rating in their own sheet of paper. After

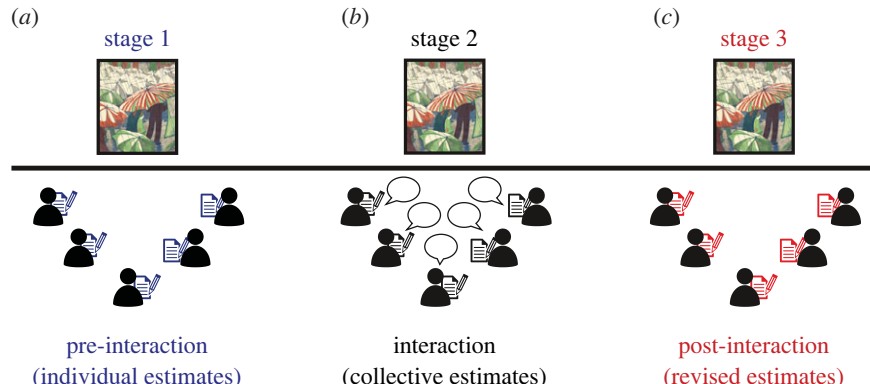

**Figure 1.** Experimental procedure. Sequence of events. The study took place in virtual meetings using Zoom (https://zoom.us/). (*a*) Individual estimates, (*b*) participants discussed their estimates in groups of five and provided collective estimates and (*c*) revised individual estimates.

the eighth painting had been shown, and before starting the second stage of the study, participants were sent a link to a form to upload their answers. Finally, research assistants verified that all individual answers had been provided before proceeding to stage 2.

In the second stage (figure 1*b*), participants turned on their microphones to discuss their estimates and reach a collective answer for the displayed painting. Frist, we randomly assigned each of the 50 participants into 10 breakout rooms using Zoom. A different research assistant joined each breakout room and proceeded with the second stage of the study. Participants were assigned a different alphabetical code (from 'A' to 'E'), and asked to change their names accordingly in Zoom. This allowed us to ask participants about the perceived expertise of other group members. We instructed participants to discuss their estimates in order to seek consensus, making clear that there was no advantage nor disadvantage associated with providing such a collective estimate (see 'Script of instructions to participants' below). If they were able to reach consensus, then all group members wrote it on their own sheet of paper. In this stage, we presented four of the eight paintings, one after the other, for a duration of 120 s each. We displayed the paintings to the participants for the duration of the entire discussion. We call these displayed items 'discussed' paintings. After the last painting had been shown and discussed, and before starting the third stage of the study, participants were sent a link to another form to upload the answers of the second stage.

In the third stage of the experiment (figure 1*c*), breakout rooms were finished and participants were all again in a single session. Participants had their microphones muted in this stage. We displayed all paintings (both the discussed and the undiscussed ones) one after another and participants were invited to revise their initial estimates and confidence ratings if they wished to change their mind. Each of the eight paintings was presented for a duration of 20 s. As in stage 1, participants wrote down their answers in a sheet of paper and then uploaded their answers. This last form asked participants to provide their answers to the third stage, to provide basic demographic information (i.e. age and gender), and to rate the perceived expertise of other people in their group. At the end of the experiment, they were asked 'How much knowledge did other people in your group seem to have about the art market?' Participants were given four alternatives: 'No Knowledge, Little Knowledge, A Moderate Amount of Knowledge, A Great Deal of Knowledge'. Every participant answered this question about each of the four other individuals in their group (who were identified by the alphabetical code previously assigned to each of them, from participant 'A' to 'E').

## 2.4. Script of instructions to participants

We provided participants with the following instructions: 'You will be presented with eight paintings that have been selected from an online auction website in Germany. In that website, there is a price associated with each of these paintings and your task is to estimate those prices (in USD) as accurately as possible and regardless of whether or not you like the paintings. At the end of the experiment, we will tell you the correct answers. The experiment has three stages. In the first stage, you will estimate the eight prices and provide a confidence rating from 0 to 10, where 0 means 'I'm completely unsure about my estimate' and 10 means 'I'm completely sure about my estimate'. In the

second stage, you will be shown only four of these paintings and will discuss your estimates as a group. For each painting, you will have two minutes to try reaching a consensus about the prices stated in the website. There are no advantages nor disadvantages associated with reaching a collective estimate. If you were able to reach consensus, all of you need to write down the same number. If you do not reach consensus in 2 min, you should simply write down an 'X' and continue with the experiment. Finally, in stage 3, you will be shown the eight paintings and have an opportunity to revise your individual estimates and confidence ratings. There are no advantages nor disadvantages associated with revising your answers'.

## 2.5. Data exclusion criteria

We performed a series of pre-registered data exclusion procedures which can be found in https://osf.io/s89w4. We planned to exclude all data coming from groups in which at least one participant chose to retire from the experiment. We also planned to exclude from the analysis all groups in which there was no diversity of opinions on at least one painting. This is because we measured diversity using log-variances (see equation (2.1) in **Proposed analyses**) and this quantity is not well-defined when all individuals provide exactly the same price estimate. However, these two situations did not occur in the main study.

As planned, we rejected from the analysis any group that contained one or more individuals providing outlying price estimates. To define outliers, we used a non-parametric method [28] based on the distribution of average price estimates provided in stage 1. We defined outliers as those participants who deviated from the median in more than three mean absolute deviances (*mad*s). Because we did not know in advance the proportion of outliers that we would observe in the main study, nor their distribution across different quintets, we estimated such value for our pre-registration using pilot data. These exclusion criteria led to a larger sample size than the one estimated based only on the proposed analyses (see **Pilot data** for more information). In practice, we excluded 11 groups following this criterion.

We also discarded data from groups that were not able to reach consensus on all paintings. While this situation did not occur in the pilot experiment (see below) nor in a similar previous experiment in a laboratory setting (control experiment in [26]), we observed and excluded 13 groups from the main study following this procedure. After performing all pre-registered data exclusion procedures our effective sample size was $N = 570$ individuals across 114 groups.

## 2.6. Control experiment

We performed a control experiment with a non-collective task. This allowed us to estimate the correlation between diversity and herding in a context with no social influence (see H6 below for more details). Critically, if some of the effects in the main experiment were coming from a regression towards the mean from stage 1 to stage 3, we could expect to observe the same result in this control experiment. $N = 100$ participants were presented with the paintings, then performed an unrelated task (i.e. answering four general-knowledge questions randomly sampled from the questions asked in reference [26]), and finally provided revised ratings. With this set-up, we then created 'virtual quintets' by randomly sampling five different people (each time with replacement).

## 2.7. Hypotheses and statistical analyses

The first three hypotheses can be regarded as positive controls that our task produces herding behaviour (H1), increases in confidence (H2) and that both effects are modulated by discussion (H3). Based on conflicting predictions thrown by different theoretical work, we tested two alternative hypotheses about the relationship between diversity and herding. One of these hypotheses (H4a) is that groups with more diverse estimates should herd more. We also consider the alternative hypothesis that groups with more diverse estimates should actually herd less (H4b). We also hypothesized that groups with more homogeneous estimates should boost their confidence more (H5). Finally, we tested the hypothesis that the correlation between herding and diversity is driven by social interaction and compared the correlation observed in the main experiment with that same correlation in the control experiment, which lacks social influence (H6). Tables S2 and S3 in the electronic supplementary material summarize these hypotheses and display all relevant variables. Below, we describe them in further detail.

**Hypothesis 1 (H1).** <u>Group discussions should lead to herding behaviour</u>. *The log-variance of price estimates for discussed paintings in stage 3 should be lower than in stage 1.*

To test this hypothesis, we estimated the log-variance of price estimates for all paintings in stages 1 and 3 of our experiment (figure 1). Given a group $j$ that provided a set of price estimates for painting $k$ at stage $s$, we may represent those values using a vector $x_{jk}^{(s)}$ of length $n$, where $n$ is the group size.

We define $LV_{jk}^{(s)}$ as

$$LV_{jk}^{(s)} = \log_{10}\left(\frac{1}{n-1}\sum_{i=1}^{n}(x_{ijk}^{(s)} - \bar{x}_{jk}^{(s)})^2\right), \tag{2.1}$$

where $x_{ijk}^{(s)}$ is the $i$th element (corresponding to the $i$th individual in group j) of vector $x_{jk}^{(s)}$, $\bar{x}_{jk}^{(s)}$ is its mean, and $n = 5$ in our experiment. By log-transforming variance in equation (2.1), we ensure that $LV_{jk}^{(s)}$ is an unbounded quantity and thus does not suffer from 'floor effects' (see **H4** below for further details).

Grounded on previous theoretical efforts across different fields, we defined 'herding' as the 'convergence of beliefs in a group through local interactions' [1,2,29,30]. If opinions converge as a result of interaction, we should observe a reduction in variance of opinions in the group. To evaluate if variance indeed decreased after discussion, we averaged $LV_{jk}^{(s)}$ across all discussed paintings separately for stages 1 and 3. If $\langle d \rangle$ represents the indexes corresponding to the discussed paintings, we measured the mean log-variance for discussed paintings (LVD) at stage $s$ and group $j$ by computing

$$LVD_j^{(s)} = \frac{1}{n_d}\sum_{k\in\langle d\rangle} LV_{jk}^{(s)}, \tag{2.2}$$

where $n_d = 4$ is the number of discussed paintings.

We hypothesized that the variable LVD would be lower at stage 3 compared to stage 1. To formally test this hypothesis, we performed a two-sided Wilcoxon signed-rank test for equal medians, which is non-parametric and does not rely on the assumption that log-variances follow any particular distribution.

**Hypothesis 2 (H2).** <u>Group discussions should boost confidence</u>. *Confidence ratings for discussed paintings in stage 3 should be higher than in stage 1.*

We performed a similar analysis to the one described for **H1** and evaluated if group discussions boosted confidence. Given a group $j$ that provided a set of confidence ratings for painting $k$ at stage $s$, we may represent those values using a vector $c_{jk}^{(s)}$ of length $n = 5$. We defined the mean confidence for discussed paintings (CD) at stage $s$ and group $j$ as

$$CD_j^{(s)} = \frac{1}{n_d}\sum_{k\in\langle d\rangle} \bar{c}_{jk}^{(s)}, \tag{2.3}$$

where $\bar{c}_{jk}^{(s)}$ is the mean of vector $c_{jk}^{(s)}$.

We hypothesized that CD would increase between stages 1 and 3. To formally test this hypothesis, we performed a two-sided Wilcoxon signed-rank test and set a criterion to reject the null hypothesis of equal medians using a 5% significance level.

**Hypothesis 3 (H3)**. <u>We should observe more herding and a larger boost of confidence in discussed compared to undiscussed paintings</u>. *The reduction in variance (**H1**) and the increase in confidence (**H2**) between stages 1 and 3 should be more pronounced in discussed paintings than in undiscussed paintings.*

To quantify herding, we defined a variable $H$ as the mean negative change in the log-variance of price estimate for group $j$ and painting $k$, i.e.

$$H_{jk} = LV_{jk}^{(1)} - LV_{jk}^{(3)}. \tag{2.4}$$

This definition ensures that more herding means a greater reduction in variance. We quantified the amount of herding observed in group $j$ as the average of $H_{jk}$ across discussed items ($HD_j$) and compared it with its average across undiscussed items ($HU_j$). We predicted that $HD_j$ would be larger than $HU_j$ (**H3a**). To formally test this hypothesis, we pre-registered a one-sided Wilcoxon signed-rank test and set a criterion to reject the null hypothesis of equal medians using a 5% significance level. To confirm that the result holds after performing a more conservative analysis, we also ran a two-sided test which was not pre-registered.

Similarly, we measured the change in mean confidence between stages 1 and 3 ($\Delta C$) for group $j$ and painting $k$, i.e.

$$\Delta C_{jk} = \bar{c}_{jk}^{(3)} - \bar{c}_{jk}^{(1)}. \tag{2.5}$$

We then averaged this quantity across discussed items ($\Delta CD_j$) and compared it with its average across undiscussed items ($\Delta CU_j$). We predicted that $\Delta CD_j$ would be larger than $\Delta CU_j$ (**H3b**). To formally test this hypothesis, we performed a two-sided Wilcoxon signed-rank test and set a criterion to reject the null hypothesis of equal medians using a 5% significance level.

**Hypotheses 4a and 4b (H4a and H4b)**. <u>Groups with more diverse estimates should herd differently.</u> *The decrease in log-variance of price estimates should be higher (H4a) or lower (H4b) for groups with more diversity in their initial opinions.*

Our main goal was to empirically test whether groups holding more diverse beliefs herd more or less than groups holding similar opinions. These hypotheses are grounded on conflicting predictions from different theoretical research across diverse fields (see Introduction for more details).

In principle, one potential issue with our proposed analysis is that it could be subject to a 'floor effect'. Low-variance groups might show less reduction in variance simply because they have less room to reduce it than high-variance groups. To rule out this interpretation, we used log-variances (equation (2.1)) which are unbounded quantities and do not suffer from 'floor effects'. Reductions in log-variance are mathematically equivalent to taking the log ratio of variances. To visualize how different linear reductions in log-variance map onto actual reductions in variance, see electronic supplementary material, figure S3.

We quantified herding using equation (2.4) and averaging across discussed paintings ($HD_j$). To quantify the diversity in initial opinions we also used log-variances but, critically, using only undiscussed paintings at stage 1. This is to prevent our analysis being influenced by regression to the mean. In other words, one could expect that groups with greater variance at stage 1 will regress towards the mean and show a relatively lower value at stage 3, simply due to measurement noise or other sources of randomness. In principle, this could lead to the misleading interpretation that more diverse groups herd more, even in the absence of such effect. To avoid this potential caveat in our analysis, we used a different set of paintings to compute herding and diversity. Since these two variables are measured on a disjoint set of items, one can be sure that regression to the mean does not play a role in our analysis. Thus, the variable 'diversity' ($D$) of group $j$ is defined as

$$D_j = LVU_j^{(1)} = \frac{1}{n_u} \sum_{k \in \langle u \rangle} LV_{jk}^{(1)}, \tag{2.6}$$

where $\langle u \rangle$ represents the indexes corresponding to the undiscussed paintings and $n_u = 4$ is the number of undiscussed items. Computing diversity using a different set of items to the ones used to compute herding is justified only if $D$ is a stable property of groups, namely if mean log-variances across undiscussed items ($LVU_j^{(1)}$) are correlated with mean log-variances across discussed items ($LVD_j^{(1)}$). Data from our pilot experiment suggests that this is the case (see **Pilot Data** for details).

Our two alternative hypotheses can be quantitatively tested by looking at the correlation between variables $D_j$ and $HD_j$. A significantly positive correlation would suggest evidence in favour of H4a (greater herding in more diverse groups) and a negative correlation would indicate evidence for H4b (less herding in more diverse groups). To avoid relying on normality assumptions about the distribution of $D_j$ and $HD_j$, we measured the Spearman rank correlation coefficient between them. To reject the null hypothesis, we used a two-sided 5% significance level.

**Hypothesis 5 (H5)**. <u>Groups with more homogeneous estimates should boost their confidence more.</u> *The increase in confidence between stages 1 and 3 should be higher for less diverse groups.*

To study how confidence is modulated by the diversity of initial opinions, we correlated $D_j$ with the mean change in confidence rating across discussed paintings ($\Delta CD_j$). The latter variable was computed using equation (2.5) and averaging variable $\Delta C$ across discussed items.

We note that measuring diversity ($D_j$) using undiscussed items is unnecessary here given that the concern about regression to the mean explaining our findings does not apply to **H5** (because we correlate variance with mean confidence). However, to be consistent in our definition of diversity across all hypotheses, we estimated diversity using equation (2.6).

Based on previous studies showing that social agreement boosts confidence (see Introduction for details), we predicted that more homogeneous groups (with lower values of $D_j$) should show a greater increase in confidence (higher values of $\Delta CD_j$). In other words, we hypothesized a negative

correlation between $D_j$ and $\Delta CD_j$. To test this hypothesis, we measured the Spearman correlation between these two variables and used a two-sided 5% significance level to reject the null hypothesis.

**Hypothesis 6 (H6)**. <u>The correlation between herding and diversity should be driven by social influence</u>. *Once the sign of the correlation between herding and diversity (H4) under social influence (main experiment) is established, its unsigned value should be larger than the unsigned correlation between those same variables in a control non-collective task (control experiment).* We computed the correlation between variables $D_j$ and $HD_j$ for different sets of 650 randomly selected virtual quintets obtained in the control experiment. To increase the robustness and representativeness of this analysis, we performed the random selection of 650 virtual quintets 1000 times. We then computed the correlation between $D_j$ and $HD_j$ for each of these 1000 sets of virtual quintets and compared the real data with the set yielding median correlation between those two variables. We performed a two-sided test for equal correlation coefficients for independent samples using a 5% significance level.

## 2.8. Secondary analyses

To test whether participants paid attention to stage 1 responses, we compared the accuracy (average individual error) and variability (average log-variance) of discussed and undiscussed items. These analyses could in principle be useful to flag the possibility that stage 1 responses were produced under lack of attention if we found that either of these two quantities is significantly different across the discussed and undiscussed conditions (two-sided paired $t$-test, 5% significance level).

We also evaluated whether herding behaviour depended on differences in perceived expertise. If that were the case, we reasoned that there should be more herding in groups where there are larger gaps in perceived expertise. To quantify this intuition, we computed for each individual a score of 'projected expertise' by averaging the expertise ratings provided by all other group members. Then, we computed the variance of projected expertise for that group (with higher values indicating a larger gap in perceived expertise) and correlated that quantity with $HD_j$ (equation (2.4)). We finally measured the Spearman correlation between these two variables and used a two-sided 5% significance level to evaluate whether it is possible to reject the null hypothesis that perceived expertise does not play a role in herding behaviour.

## 2.9. Pilot experiment

We performed a pilot experiment to estimate the sample size needed to test our hypotheses. Nineteen groups of five individuals participated in our pilot study. Participants were visitors to the TATE Exchange exhibition at the TATE Modern gallery in London ($N = 95$, 54 female, mean age: 35.15 years, range: 18–67 years). All participants were naive, had normal or corrected-to-normal vision and provided written informed consent. We tested one group at a time. The only difference between this and the main study is that interactions in the pilot experiment were face-to-face rather than virtual interactions through Zoom.

Using the pilot data, we performed the three positive controls explained in hypotheses **H1–H3**. We found that discussions led to a significant reduction in variance (median $LVD_j^{(1)} = 10.1$, median $LVD_j^{(3)} = 9.0$, $z = 3.1$, $p = 0.001$), and a significant boost in confidence (median $CD_j^{(1)} = 2.8$, median $CD_j^{(3)} = 3.4$, $z = 3.1$, $p = 0.001$). Consistent with **H3a**, we found that there was significantly more herding in discussed compared to undiscussed paintings (median $HD_j = 0.55$, median $HU_j = 0.10$, $z = 2.4$, $p = 0.01$). The increase in confidence was larger in discussed compared to undiscussed paintings (H3b), but this effect did not reach significance (median $\Delta CD_j = 1.1$, median $\Delta CU_j = 0.50$, $z = 1.5$, $p = 0.07$).

## 2.10. Estimation of sample size

We estimated the sample size needed to test our five hypotheses. Based on the power analysis described below, we found that the critical hypothesis is **H4** (i.e. that we will observe a positive correlation between $D_j$ and $HD_j$) given that it requires the largest sample size (see electronic supplementary material, figure S2 for details about the power analysis performed for each hypothesis separately).

In our pilot experiment, we observed a positive correlation between $D_j$ and $HD_j$ that did not reach significance ($r_s = 0.31$, $p = 0.10$). Based on this data, we estimated the sample size using a resampling approach: we created sets of surrogated data with different sample sizes by randomly resampling with replacement our pilot data. Using this method, we simulated sample sizes ranging from 10 to 120 groups (i.e. $N = 50$ to $N = 600$ participants) in steps of 10 groups and repeated this procedure 10 000 times for each sample size. To compute the expected power at each sample size, we measured

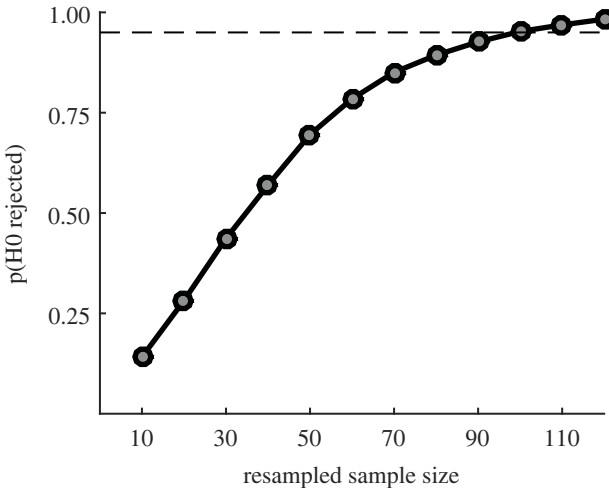

**Figure 2.** Power analysis. Probability to reject the null hypothesis in H4 as a function of resampled sample size (i.e. number of quintets). Power was estimated by computing the fraction of samples (out of 10 000) where we observed a significant result in our pilot data at 5% significance level.

the fraction of samples that produced a significant result with a two-sided 5% significance level. We observed that in order to achieve a 95% power, we would need data from $N = 600$ participants across 120 groups (figure 2).

Regarding Hypothesis 5, we observed a trend towards observing a negative correlation between $D_j$ and $\Delta CD_j$ ($r_s = -0.37$, $p = 0.06$). The same resampling approach used to estimate the sample size in figure 2 suggested that we would need at least 100 groups to detect this effect with 95% power (see electronic supplementary material, figure S2 for more details).

## 2.11. Stability of diversity across discussed and undiscussed items

To justify our measurement of diversity using undiscussed items and correlating this variable with herding behaviour (equation (2.4)) and changes in confidence (equation (2.5)) for discussed items, it is necessary to show that diversity is a stable property of groups. To evaluate this, we computed $D_j$ as in equation (2.6) by taking the mean log-variance across undiscussed items ($\text{LVU}_j^{(1)}$) and tested if it was correlated with the mean log-variance across discussed items ($\text{LVD}_j^{(1)}$). This was done using the data collected in the pilot study. We found a significant correlation between them ($r_s = 0.82$, $p = 1 \times 10^{-5}$), which suggests that our metric of diversity is stable across the two different sets of paintings.

## 2.12. Correction of sample size based on exclusion criteria

Using the pilot data, we also estimated the fraction of groups that we expected to discard based on our data exclusion criteria. We observed that less than 4% of our participants (3.16%) were considered outliers. We also discarded data from zero-variance groups. Based on the observation that no groups had zero variance in the pilot experiment and that less than 1% of them had zero variance in a large-scale previous experiment [26], we estimated an upper bound for this situation of 1%. Setting a worst-case scenario of 5% of outliers or zero-variance groups in the main study, and assuming that those participants were distributed across different groups, we increased sample size by five additional groups in order to achieve a statistical comparison with 95% power. Overall, using this correction, we collected data from $N = 650$ participants across 130 groups. After performing all our pre-registered data exclusion procedures, the effective sample size was $N = 570$ participants across 114 groups (for details, see 'Data exclusion criteria' above), yielding over 95% power for all hypotheses (figure 2; electronic supplementary material, figure S2).

## 2.13. Comparison of price estimates and other general-knowledge questions

Our main objective is measuring the correlation between group diversity and herding behaviour. In this regard, the price-estimation task was selected because it induced a large heterogeneity in group variances, as confirmed with the pilot data (pre-interaction within-group log-variances, mean ± s.d. =

9.63 ± 1.96). However, one may very well wonder how this compares to a more classical setting where participants are asked to estimate general-knowledge quantities such as the weight of cattle, the height of monuments, or the length of rivers. To answer this question, we compared the measured group variances in the pilot experiment with the ones induced in a previous study [26]. We found that the group variances observed in that study (pre-interaction within-group log-variances, mean ± s.d. = 3.58 ± 1.41) were significantly smaller than the one measured in the pilot experiment (rank sum test, $z = 14.2$, $p = 8 \times 10^{-46}$). We also observed that the heterogeneity in pre-interaction within-group log-variances in the previous study was also smaller than in the pilot data (Fisher test for equal variances, $F_{83975} = 0.53$, $p = 1.7 \times 10^{-5}$). Overall, this analysis suggested that the price-estimation task proposed in this study was better suited to address our hypotheses compared to a more classical setting based on general-knowledge questions.

## 2.14. Code and data availability

All codes and data supporting our estimation of sample size using pilot data are available for download at: https://figshare.com/s/924ff1880973b5da842a. All raw data and codes associated with the main and control preregistered studies are available at the Open Science Framework (https://osf.io/n5zd6/).

# 3. Results

## 3.1. Preregistered main analyses

Following our pre-registered protocol (https://osf.io/s89w4/), we first tested the hypothesis that group discussions should lead to herding (H1). We observed that the log-variance of price estimates for discussed paintings at stage 3 were significantly lower than in stage 1 (figure 3a; Cohen's $d = 0.55$, $z = 5.6$, $p = 2 \times 10^{-8}$), suggesting that the selected task successfully produced herding behaviour. Also, and in line with our second hypothesis (H2), we observed that confidence ratings for discussed paintings in stage 3 were significantly larger than in stage 1 (figure 3b; Cohen's $d = 1.22$, $z = 8.5$, $p = 2 \times 10^{-17}$).

Our third hypothesis was that we should observe more herding (H3a) and a larger boost of confidence (H3b) in discussed compared to undiscussed paintings. In line with hypothesis H3a, we found that herding (equation (2.4)) for discussed items was significantly larger than for undiscussed paintings (figure 3c; Cohen's $d = 0.23$, $z = 3.0$, $p = 0.001$). This result suggests that group discussions promoted herding behaviour. However, and against hypothesis H3b, we did not find any evidence that the boost in confidence for discussed paintings was larger than the one observed for undiscussed items (figure 3d; Cohen's $d = 0.09$, $z = 0.07$, $p = 0.51$). This observation indicates that the increase in confidence for discussed items may not be driven solely by actual deliberation.

The main aim of this study was to test whether there is a positive (H4a) or negative (H4b) correlation between diversity (equation (2.6)) and herding. We observed a significantly positive correlation between these two variables (figure 4a; Spearman correlation $r_s = 0.33$, $p = 4 \times 10^{-4}$), which indicates that more diverse groups herded more. Our fifth hypothesis was that the increase in confidence for discussed paintings should be greater for more homogeneous groups (H5). We tested this hypothesis by looking at the correlation between the boost in confidence and diversity. However, we did not observe any evidence in support of H5 (figure 4b; Spearman correlation $r_s = -0.06$, $p = 0.51$) i.e. that changes in confidence following social influence were influenced by the diversity of opinions in the group.

Finally, we asked whether the correlation between herding and diversity depended on social influence (H6). To study this, we performed a control study with no social influence (see Methods for details). With these data, we created virtual quintets that allowed us to recompute the correlation between herding and diversity across 1000 datasets, all with the same sample size as the main study (see Methods for details). In line with H6, we found that the correlation between diversity and herding in the main study was significantly larger than the one yielding the median value of correlation coefficient in the control study (figure 4c; test for equal correlations of independent samples, $z = 2.2$, $p = 0.01$). This analysis provides evidence that the observations supporting H4a were driven by social influence.

## 3.2. Preregistered secondary analyses

We pre-registered two secondary analyses in this work. The first one sought to compare the accuracy and variability of stage 1 responses across discussed and undiscussed paintings. We observed no evidence

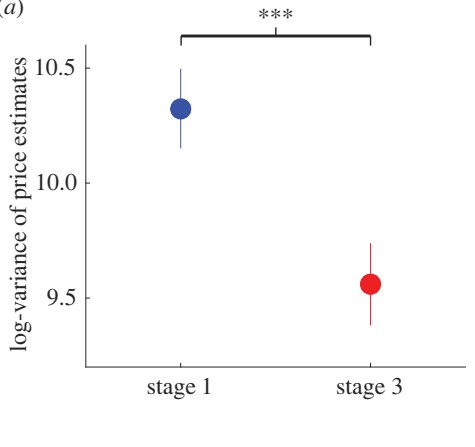

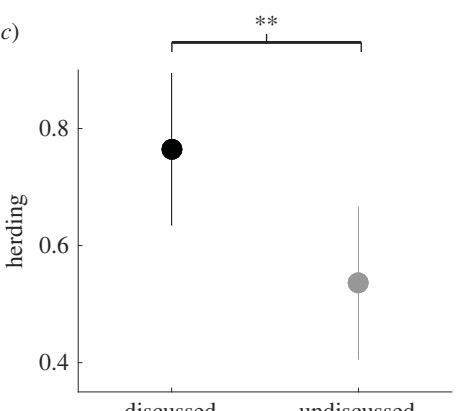

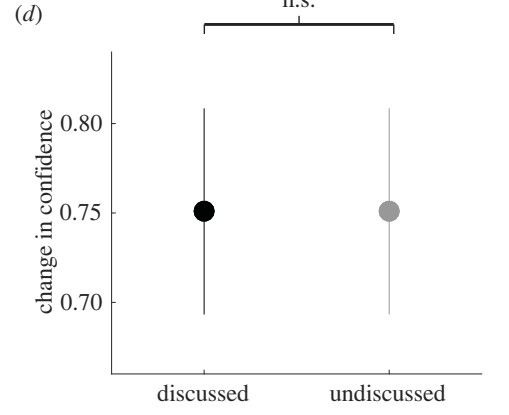

**Figure 3.** Herding behaviour and boost in confidence following social influence. (*a*) The log-variance of price estimates decreased from stage 1 to stage 3. (*b*) Mean confidence ratings increased from stage 1 to stage 3. (*c*) Herding, defined in equation (2.4), was more pronounced in discussed paintings compared to undiscussed paintings. (*d*) The change in confidence was approximately equal for discussed and undiscussed paintings. In all panels, dots represent mean and vertical lines depict SEM. Blue: data from stage 1. Red: data from stage 3. Black: data from discussed paintings. Grey: data from undiscussed paintings. n.s.: $p > 0.05$, $^{**}p < 0.01$, $^{***}p < 0.001$.

that the accuracy (Cohen's $d = 0.24$, unpaired *t*-test, $t_{226} = 1.8$, $p = 0.07$) or variability (Cohen's $d = 0.14$, unpaired *t*-test, $t_{226} = 1.1$, $p = 0.28$) of the responses provided at stage 1 was different across discussed versus undiscussed items.

The other pre-registered secondary analysis evaluated the potential role of perceived expertise in herding behaviour. We studied whether there was more herding in groups with larger gaps in perceived expertise. To this end, we computed for each individual a score of 'projected expertise' by averaging the expertise ratings provided by all other group members. Then, we computed the variance of projected expertise for that group and looked at its correlation with herding. We found that there was no significant correlation between variance in projected expertise and herding (Spearman correlation $r_s = -0.05$, $p = 0.60$), suggesting that perceptions of expertise played a minor or no role in promoting herding behaviour.

## 3.3. Non-preregistered analyses

We asked whether this task led to herding behaviour for non-discussed items. This could happen, for example, if participants learned through social influence that they generally tend to underestimate or overestimate other people's values, and so they may adjust their responses in stage 3, even for items that remained undiscussed. To examine this possibility, we performed a non-preregistered analysis where we compared the mean log-variance of price estimates at stages 1 and 3 across non-discussed paintings. Running the same statistical test as in H1 (but for non-discussed paintings), we found a significant reduction in variance (Cohen's $d = 0.38$, $z = 3.8$, $p = 0.0002$). This result suggests that social

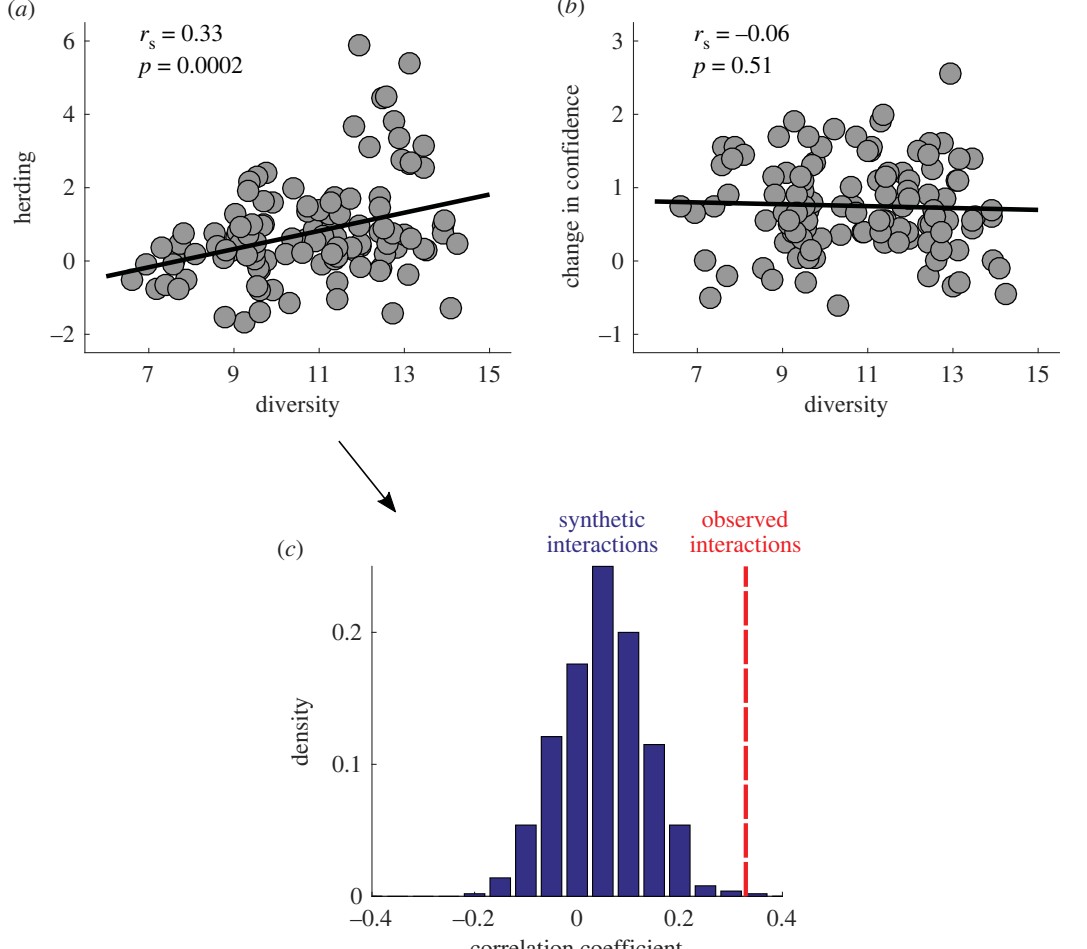

**Figure 4.** Diversity of beliefs, herding behaviour, and confidence. (*a*) Herding correlated positively with diversity. (*b*) We observed no correlation between change in confidence and diversity. In these panels, each dot represents a group, black lines show the best fitting linear regression, and we display in the Spearman correlation coefficients ($r_s$) and its *p*-value. (*c*) Using the data obtained in the control study, we estimated the correlation between diversity and herding in the absence of social influence. Blue bars show the probability density of those correlation coefficients obtained without social interaction. The red dashed line shows the observed correlation coefficient in the main study (*a*).

influence may have promoted herding behaviour even for beliefs that were not explicitly interchanged. Similarly, we asked whether individuals changed their confidence in paintings that were not discussed and found a significant increase in ratings from stage 1 to stage 3 (Cohen's $d = 1.1$, z = 8.3, $p = 2 \times 10^{-16}$).

In the main study, we found that herding was more pronounced in discussed compared to undiscussed items (figure 3*c*) using a one-sided Wilcoxon sign rank test, as pre-registered. After following the protocol, we confirmed, in a non-preregistered analysis, that this result holds after performing the two-sided version of the same statistical test (z = 2.9, $p = 0.002$).

When we tested Hypothesis 6, we compared the correlation between herding and diversity in the main study with the one yielding median correlation in the distribution displayed in figure 4*c*. This preregistered analysis evaluated whether the correlation between these variables in the main study was larger than the one that would be typically observed in the absence of social influence. However, an alternative way to analyse these data is by looking at the fraction of samples (i.e. virtual experiments with no social influence) that produce a larger correlation than the one we actually observed. Indeed, this fraction could be interpreted as a *p*-value: it reflects the proportion of times where we find a more extreme result than the one we empirically observed under the null hypothesis of no social influence. We found that only 3 out of 10 000 simulations produced a larger correlation than the one found in the main study ($p = 0.0003$). This non-preregistered reanalysis of the data provides further support that the correlation between herding and diversity relies on social influence.

Finally, we studied whether the effect of diversity would still be present if the variable 'herding' (equation (2.4)) was binary. We therefore coded herding as 0 if individuals did not converge in their

estimates or even diverged from each other, and set a value of 1 if there was any herding at all (overall, 77.1% of the groups showed some non-zero value of herding). A logistic regression revealed that diversity still significantly modulated herding (best-fitting estimate: $\beta = 0.29 \pm 0.13$, $t_{112} = 2.3$, $p = 0.02$) even when this variable was coded as binary. This analysis rules out the possible concern that larger values of herding in diverse groups are because individuals have more room to herd (for more information, see also electronic supplementary material, figure S3).

# 4. Discussion

Our findings show that herding was larger in groups holding increasingly varied opinions. This result promotes our understanding of social alignment, by combining three factors which occur in many social situations: (1) high uncertainty about the accuracy of one's opinion (known to increase reliance on social information, see [31]); (2) a lack of a clear majority view; and (3) the non-trivial task of reaching agreement with others. Here, rather than faking [10,11] a socially dominant view or explicitly displaying the average of others' opinions [9], participants were exposed to naturally occurring statistically varied opinions present in everyday interactions. Though all participants were provided with the same instructions to reach a non-binding consensus at the end of the discussion, and were all influenced to change their mind by this process of deliberation, herding behaviour depended on the crowd statistics, more specifically, on the variance of group beliefs.

Our task mimics many characteristics of social conversations and exchange of opinions—it combines high subjective uncertainty about what to believe as well as about who to follow. These features allowed us to see that participants responded differently to different distributions of uncertain opinions. However, while several theories would have predicted that groups with a lesser initial diversity of opinions would be closer to exhibiting a cohesive group-view and therefore foster more herding, we observed more herding in groups with wider diversity: less reliable social information produced more alignment among group members. Our data are then at odds with research showing that incidental similarity leads to stronger social influence [32].

Here, we focused on an estimation task where non-expert opinions are highly uncertain. Initial beliefs are not based on any prior knowledge or learned skill, but they probably reflect subjective opinions that arise from previous experiences and various rules of thumb. One explanation to these findings is that these beliefs might have been transient, weak or not convincing enough to induce any stress [33] or potential damage to self-esteem or reputation [34] when participants found themselves disagreeing with others in the most similar groups. Conversely, observing the radical divergence of viewpoints in the more diverse groups may have induced sufficient reputational and self-esteem concerns in participants to push towards converging with others. Further research is needed to test whether affective components played a meaningful role in this set-up.

An alternative explanation may be related to the informational role provided by crowd statistics. Diversity could mean that peers drew information from independent sources, so combining private and social signals might indeed be more appropriate in high-variance groups [19]. In these terms, interacting with others who believed in price estimates in a completely different order of magnitude might also increase the uncertainty about the validity of one's initial estimate and led to herding. Indeed, this strategy—known as 'copy when uncertain'—has previously shown to have adaptive value [35]. In addition, holding low-quality private information has been shown to boost the importance of social information in collective systems [36]. Conversely, interacting with people who hold similar beliefs might act as confirmatory evidence and reduce subjective uncertainty [37].

One limitation of this study is that participants' estimates about the market value of paintings could have been influenced by individuals' value-based judgements. While we explicitly asked participants to focus on the actual price of the painting, these data cannot rule out the possibility that estimates were partially driven by how much people liked the paintings. Therefore, it remains unknown whether these findings would be similar in tasks where value-based judgements are less relevant, such as in the estimation of the weights of objects or general-knowledge quantities. Another limitation of this study is that the observed effect of diversity on herding may strongly depend on how we defined 'diversity' (as the variance of opinions in a group). Future research should examine whether and how different kinds of diversity (e.g. in appearance, accent, profession, personality, cognitive style etc.) influence the convergence of opinions in group settings.

Taking a broader perspective, the impact of diversity on opinion dynamics opens a series of important questions on the epistemic benefits of allowing or nurturing a diversity of opinions [38],

but also on its political implications. Our findings here resurrect the observations first made by one of the founding fathers of modern political science. During his visit to North America in 1831, Alexis de Tocqueville noted that democratic societies, which stimulate the flourishing of diverse and independent attitudes, paradoxically led to a highly conformist society, finally producing a narrow range of individual tastes and opinions [17,18]. What could sound like a paradox under the pen of Tocqueville is in line with our results: under uncertainty, crowd statistics differently influences changes of mind, and a wider diversity of views favours patterns of herding or, as Tocqueville and others after him preferred to call it, more conformism.

Ethics. All protocols proposed in this study were approved by the ethics committee at CEMIC (Centro de Educacion Medica e Investigaciones Clinicas 'Norberto Quirno', Buenos Aires, Argentina) protocol no. 435-5.

Data accessibility. All codes and data supporting our estimation of sample size using pilot data are available for download at: https://figshare.com/s/924ff1880973b5da842a. All raw data and codes associated with the main and control preregistered studies are available at the Open Science Framework (https://osf.io/n5zd6/).

Electronic supplementary material is available online [39].

Authors' contributions. J.N.: conceptualization, data curation, formal analysis, funding acquisition, investigation, methodology, project administration, resources, software, supervision, validation, visualization, writing—original draft, writing—review and editing; O.A.: conceptualization, investigation, methodology, writing—review and editing; R.M.: conceptualization, methodology, writing—original draft, writing—review and editing; B.B.: conceptualization, investigation, methodology, project administration, resources, supervision, writing—original draft, writing—review and editing; O.D.: conceptualization, funding acquisition, investigation, methodology, project administration, resources, supervision, writing—original draft, writing—review and editing.

All authors gave final approval for publication and agreed to be held accountable for the work performed therein.

Conflict of interest declaration. We declare we have no competing interests.

Funding. J.N. was supported by the James McDonnell Foundation Twenty-first Century Science Initiative in Understanding Human Cognition—Scholar Award (grant no. 220020334) and by a Sponsored Research Agreement between Facebook and Fundación Universidad Torcuato Di Tella (grant no. INB2376941). B.B. and O.D. were supported by the research grant 'Diversity in Social Environments' (DISE) of the Nomis Foundation.

Acknowledgements. We would like to thank Julieta Ruiz, Eugenia Albornoz, Mauricio Schumacher, Luciana Uhrich, Milagro Urricariet, Candela Jantus, Ludmila Lemme, Mercedes Schiappapietra, Federico Barrera Lemarchand, Micaela Cabo, Ivan Caro, Catalina de Lafuente, Martin Winter, Nicolás Laveglia, Florencia Bustos, and Joaquin Gonzalez for assistance with data collection.

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
