## [Peer Review File · Royal Society Open Science]

Review History

RSOS-191497.R0 (Original submission)

Review form: Reviewer 1 (John Michael)

Do you have any ethical concerns with this paper?

No

Recommendation?

Accept in principle

Comments to the Author(s)

This is a very interesting and well-designed study.

The hypotheses are well-motivated and clearly stated, and the design is adequate to test them. The planned analyses seem appropriate and are stated with sufficient clarity and precision to limit 'undisclosed flexibility'. The sample size is justified. The methodology is also sufficient clear and detailed to enable replication.

Here are a few observations that may be useful to consider before finalizing the design:

- One suggestion with respect to the idea that when people's opinions are close to each other they may assume they are relying on a common source of information, so they should have less reason to herd: Would you expect this to be influenced by what they know about the commonness or lack thereof of the source of info? In the current design of course they know that they all have seen the same paintings, but their beliefs about others' expertise may play a role. For example, if your estimate is wildly different from mine and I think you seem to have more expertise (e.g. because you have indicated this or because you appear confident), I may adjust to match you, but perhaps not if I think we have the same level of expertise. This is first of all just a general question and need not be taken into account here, but it could be a point for the discussion or for an experiment 2 (e.g. manipulating their beliefs about others' expertise or confidence). Or perhaps you could probe this by asking them how confident or competent the others in their group seemed.

- In stage 2, what if they cannot come to a consensus?

- In stage 2, what if they write down different sums rather than the one collectively agreed?

- In stage 3, 'participants will be invited to revise their initial estimates'. Does this prime them to revise/herd? (Even just using the word 'initial' may encourage revision.

- Generally, I would worry a bit that they may not take stage 1 seriously, since they know they will have plenty of time to consider their response later and to take into account what they have heard from the others. It may be worth checking this for example by testing whether stage 1 responses in this design are less variable (with people just putting down a stock response each time) than in a treatment where there is only stage 1, or where they are told in stage 1 which paintings will be discussed with the group in stage 2. Or comparing to data from a previous experiment if you do not want to run this extra treatment. To be clear: I am not insisting that this is necessary; it is just a suggestion.

- Will the stage 2 discussions be recorded? It could be interesting to see if there are some features of these interactions which predict herding and/or confidence boosts (The work by Fusaroli and Tuyen and colleagues could provide some ideas here). It could also be useful to know if some of them take on the role of experts and influence the others in the group (perhaps because they really are more knowledgeable or perhaps just because they are more confident)

Review form: Reviewer 2

Do you have any ethical concerns with this paper?

No

Recommendation?

Reject

Comments to the Author(s)

The manuscript "Diversity of opinions promotes herding in uncertain crowds" describes a quasi-experimental design to test whether initial high variance in opinions will lead to more private conformity after group decision. While I was rather enthused by the description of the approach in the abstract and the question how variance in opinion may alter social influence, reading through the manuscript vastly dampened my enthusiasm: The introduction lacks the necessary rigor and ignores most classic literature and the presented design is unconvincing and unclear.

Introduction:

While I wonder why the authors cite Milgrim's obedience studies, as they only marginal relate to the research question. Instead, they should have considered the classic distinction between informational and normative influence (Deutsch & Gerard, 1955). This is crucial as it will introduce the differentiation between private and public conformity and makes clear that the presented study will be rather scrutinizing informational influence, where people infer knowledge from other sources while being unaware of any objective standard (Sherif, 1935), instead of normative influence, where they will maintain their private opinion on a clear, objective truth while being pressured to publicly conform with the crowd (Asch, 1956). The authors are completely ignoring this rather large part of the literature that deals with ambiguous stimuli (what the authors name uncertainty) and informational influence.

Furthermore, it appears to me that arguments for the diversity hypothesis (H4) constitute a post-hoc rationalization of trends in the pilot data. While, provided there is an exact true answer unknown to the participants, people should be cautious not to rely on low-reliability shared cues, convergence of estimates mostly indicates correct inference and not overreliance on wrong cues. This is also reflected in the author's argumentation in the following paragraph that similarity should also boost confidence. To determine an overreliance on correlated cues, one would need to observe the convergence of cues, not of estimates. A good discussion on how certainty and diverging answers can be used as cues for correct answers, please see Moussaid, Kämmer, Analytis, Neth (2013). Speaking even more profoundly against the author's rationale, in the case presented, there is no objective truth and thus no cue to follow to exploit wisdom of crowds: The monetary value of art is a very philosophical question and results of an auction in Germany, that is some while ago (and the time is not specified in the manuscript), is not a very good indicator of the current market price for these paintings in Argentina. Moreover, the authors argue that they deliberately chose a scenario, in which there is an uncertainty of truth. Alternatively, if they want to study wisdom of crowds, they should ask for estimates on subjects, for which there is an objective truth clearly unknown to participants (length of (not very famous) rivers, heights, etc.).

Design:

It is not described, how participants are expected to reach consensus and how/ by whom this info will be put forward in the discussion phase. Techniques of reaching consensus might be quite different (averaging the individual estimates/ 'expert' decision/ voting) and the results might be quite variable dependent on, for instance, order of presentation of single estimates. As such, 40s appears to be not a lot of time for discussion and reaching consensus. Furthermore, the incentivization of the participants is not described. Will they receive just a show-up fee or will they be incentivized to estimate as close as possible to the German auction?

The within-control is not fully convincing to me: I am not sure that regression to the mean effects are fully controlled for, when you take just half of the items as variance indicator: the group's characteristics will remain the same (as you could show with the high correlations). Instead, a between-group control with a non-collective task will be a much more credible control for the group effects.

I am also not sure about the log-transformation performed - this will cause the exclusion of 0 variance groups and other statistical models may account for issues of bounded values of the outcome variable.

While I appreciate that pre-test data was used and that the presented power analysis considers the group as UE, the authors should not assume one-tailed testing, as the introduction made clear that opposite hypothesis is also discussed in literature.

Review form: Reviewer 3

Do you have any ethical concerns with this paper?

No

Recommendation?

Accept with minor revision

Comments to the Author(s)

Attached (see Appendix A).

Review form: Reviewer 4

Do you have any ethical concerns with this paper?

No

Recommendation?

Major revision

Comments to the Author(s)

The authors want to test a hypothesis of "herding". They suggest that in groups with greater variance, greater decrease of variance across individual valuations will occur when other members' valuations are known, but in groups with lower variance, knowing other group members' valuations will increase confidence.

Major concerns:

1. The researchers explicitly tell groups that they must converge to a value in their discussion. This naturally will lead to a decrease in variance. This seems to me to be a major design flaw that explains away all effects the researchers might find.
2. Given the natural inclination of people to shift their valuations towards the values of others (e.g., Zaki, Schirmer, & Mitchell, 2011), I am not sure why one shouldn't expect that people's values converge when they can share their valuations in a group. Moreover, one might expect that when estimates within a group are close together (i.e., exhibiting low variance), there is naturally less of an inclination to shift one's values because one is close enough anyway. This will naturally lead to lower levels of convergence in groups with lower levels of variance at t_0 ; this is something that cannot be rectified by simply looking at ratings on a log scale, nor, if we assume stability in intragroup variance across all painting, can it be rectified by calculating diversity of the non-discussed paintings.
3. Regarding the log transformation, how will researchers deal with the possibility that all the participants write the same number at the end (resulting in taking the logarithm of zero)?
4. In this case, the authors use decrease in log-variance as a proxy for "herding", but it seems that the process they are measuring here is more "convergence" rather than herding. To demonstrate herding, researchers would need to demonstrate some form of group polarisation, which cannot be captured with their measures here.

Minor:

1. Variables: While I applaud the use of equations to specify how measures are being calculated, it sometimes feels as though the authors are using them to distract readers from the actual simplicity of the measures (i.e., averaging, variance, etc.). To reach a wider audience, I would suggest that the authors use more plain language to explain the measures they are using rather than couch them in mathematical terms.

Decision letter (RSOS-191497.R0)

29-Nov-2019

Dear Dr Navajas,

The Editors assigned to your Stage 1 Registered Report ("Diversity of opinions promotes herding in uncertain crowds") have now received comments from reviewers. We would like you to revise your paper in accordance with the referee and editors suggestions which can be found below (not including confidential reports to the Editor). Please note this decision does not guarantee eventual acceptance.

To revise your manuscript, log into <http://mc.manuscriptcentral.com/rsos> and enter your Author Centre, where you will find your manuscript title listed under "Manuscripts with Decisions." Under "Actions," click on "Create a Revision." Your manuscript number has been appended to denote a revision. Revise your manuscript and upload a new version through your Author Centre. We hope to receive your revised Stage 1 Registered Report within the next 3 months.

When submitting your revised manuscript, you must respond to the comments made by the referees and upload a file "Response to Referees" in "Section 2 - File Upload". Please use this to document how you have responded to the comments, and the adjustments you have made. In order to expedite the processing of the revised manuscript, please be as specific as possible in your response.

Once again, thank you for submitting your manuscript to Royal Society Open Science and we look forward to receiving your revision. If you have any questions at all, please do not hesitate to get in touch.

on behalf of Professor Chris Chambers (Registered Reports Editor, Royal Society Open Science)
openscience@royalsociety.org

Associate Editor Comments to Author (Professor Chris Chambers):

Four expert reviewers have now appraised the manuscript. The reviews are polarised in their categorical recommendations, with two very negative (Reviewer 2 and 4) and two more positive (Reviewer 1 and 3). Although the recommendations vary, the reviewers raise many common points that span the full range of the Stage 1 criteria, from the rationale of the research question

and hypotheses (including hypothesis 4 as a key point of contention), the degree of methodological detail, the justification of specific analysis choices (e.g. log-transformations), and potentially fatal design flaws.

For a regular article, this combination of reviews would result in outright rejection, but the advantage of the Registered Reports format is that it provides the opportunity for authors to revise their study design and resolve such concerns before they present a barrier to publication. In this case, I judge that there is sufficient potential in the design and reflected in the reviews to invite such a revision; however, the manuscript will need to be very substantially revised and will be returned to the reviewers for reappraisal.

Reviewers' Comments to the Author:

Reviewer: 1

Comments to the Author(s)

This is a very interesting and well-designed study.

The hypotheses are well-motivated and clearly stated, and the design is adequate to test them.

The planned analyses seem appropriate and are stated with sufficient clarity and precision to limit 'undisclosed flexibility'. The sample size is justified. The methodology is also sufficient clear and detailed to enable replication.

Here are a few observations that may be useful to consider before finalizing the design:

- One suggestion with respect to the idea that when people's opinions are close to each other they may assume they are relying on a common source of information, so they should have less reason to herd: Would you expect this to be influenced by what they know about the commonness or lack thereof of the source of info? In the current design of course they know that they all have seen the same paintings, but their beliefs about others' expertise may play a role. For example, if your estimate is wildly different from mine and I think you seem to have more expertise (e.g. because you have indicated this or because you appear confident), I may adjust to match you, but perhaps not if I think we have the same level of expertise. This is first of all just a general question and need not be taken into account here, but it could be a point for the discussion or for an experiment 2 (e.g. manipulating their beliefs about others' expertise or confidence). Or perhaps you could probe this by asking them how confident or competent the others in their group seemed.
- In stage 2, what if they cannot come to a consensus?
- In stage 2, what if they write down different sums rather than the one collectively agreed?
- In stage 3, 'participants will be invited to revise their initial estimates'. Does this prime them to revise/herd? (Even just using the word 'initial' may encourage revision.
- Generally, I would worry a bit that they may not take stage 1 seriously, since they know they will have plenty of time to consider their response later and to take into account what they have heard from the others. It may be worth checking this for example by testing whether stage 1 responses in this design are less variable (with people just putting down a stock response each time) than in a treatment where there is only stage 1, or where they are told in stage 1 which paintings will be discussed with the group in stage 2. Or comparing to data from a previous experiment if you do not want to run this extra treatment. To be clear: I am not insisting that this is necessary; it is just a suggestion.
- Will the stage 2 discussions be recorded? It could be interesting to see if there are some features of these interactions which predict herding and/or confidence boosts (The work by Fusaroli and Tylan and colleagues could provide some ideas here). It could also be useful to know if some of them take on the role of experts and influence the others in the group (perhaps because they really are more knowledgeable or perhaps just because they are more confident)

Reviewer: 2

Comments to the Author(s)

The manuscript "Diversity of opinions promotes herding in uncertain crowds" describes a quasi-experimental design to test whether initial high variance in opinions will lead to more private conformity after group decision. While I was rather enthused by the description of the approach in the abstract and the question how variance in opinion may alter social influence, reading through the manuscript vastly dampened my enthusiasm: The introduction lacks the necessary rigor and ignores most classic literature and the presented design is unconvincing and unclear.

Introduction:

While I wonder why the authors cite Milgrim's obedience studies, as they only marginally relate to the research question. Instead, they should have considered the classic distinction between informational and normative influence (Deutsch & Gerard, 1955). This is crucial as it will introduce the differentiation between private and public conformity and makes clear that the presented study will be rather scrutinizing informational influence, where people infer knowledge from other sources while being unaware of any objective standard (Sherif, 1935), instead of normative influence, where they will maintain their private opinion on a clear, objective truth while being pressured to publicly conform with the crowd (Asch, 1956). The authors are completely ignoring this rather large part of the literature that deals with ambiguous stimuli (what the authors name uncertainty) and informational influence.

Furthermore, it appears to me that arguments for the diversity hypothesis (H4) constitute a post-hoc rationalization of trends in the pilot data. While, provided there is an exact true answer unknown to the participants, people should be cautious not to rely on low-reliability shared cues, convergence of estimates mostly indicates correct inference and not overreliance on wrong cues. This is also reflected in the author's argumentation in the following paragraph that similarity should also boost confidence. To determine an overreliance on correlated cues, one would need to observe the convergence of cues, not of estimates. A good discussion on how certainty and diverging answers can be used as cues for correct answers, please see Moussaid, Kämmer, Analytis, Neth (2013). Speaking even more profoundly against the author's rationale, in the case presented, there is no objective truth and thus no cue to follow to exploit wisdom of crowds: The monetary value of art is a very philosophical question and results of an auction in Germany, that is some while ago (and the time is not specified in the manuscript), is not a very good indicator of the current market price for these paintings in Argentina. Moreover, the authors argue that they deliberately chose a scenario, in which there is an uncertainty of truth. Alternatively, if they want to study wisdom of crowds, they should ask for estimates on subjects, for which there is an objective truth clearly unknown to participants (length of (not very famous) rivers, heights, etc.).

Design:

It is not described, how participants are expected to reach consensus and how/ by whom this info will be put forward in the discussion phase. Techniques of reaching consensus might be quite different (averaging the individual estimates/ 'expert' decision/ voting) and the results might be quite variable dependent on, for instance, order of presentation of single estimates. As such, 40s appears to be not a lot of time for discussion and reaching consensus.

Furthermore, the incentivization of the participants is not described. Will they receive just a show-up fee or will they be incentivized to estimate as close as possible to the German auction?

The within-control is not fully convincing to me: I am not sure that regression to the mean effects are fully controlled for, when you take just half of the items as variance indicator: the group's characteristics will remain the same (as you could show with the high correlations). Instead, a between-group control with a non-collective task will be a much more credible control for the group effects.

I am also not sure about the log-transformation performed – this will cause the exclusion of 0 variance groups and other statistical models may account for issues of bounded values of the outcome variable.

While I appreciate that pre-test data was used and that the presented power analysis considers the group as UE, the authors should not assume one-tailed testing, as the introduction made clear that opposite hypothesis is also discussed in literature.

Reviewer: 3

Comments to the Author(s)

Please see the attached file.

Reviewer: 4

Comments to the Author(s)

The authors want to test a hypothesis of "herding". They suggest that in groups with greater variance, greater decrease of variance across individual valuations will occur when other members' valuations are known, but in groups with lower variance, knowing other group members' valuations will increase confidence.

Major concerns:

1. The researchers explicitly tell groups that they must converge to a value in their discussion. This naturally will lead to a decrease in variance. This seems to me to be a major design flaw that explains away all effects the researchers might find.
2. Given the natural inclination of people to shift their valuations towards the values of others (e.g., Zaki, Schirmer, & Mitchell, 2011), I am not sure why one shouldn't expect that people's values converge when they can share their valuations in a group. Moreover, one might expect that when estimates within a group are close together (i.e., exhibiting low variance), there is naturally less of an inclination to shift one's values because one is close enough anyway. This will naturally lead to lower levels of convergence in groups with lower levels of variance at t_0 ; this is something that cannot be rectified by simply looking at ratings on a log scale, nor, if we assume stability in intragroup variance across all painting, can it be rectified by calculating diversity of the non-discussed paintings.
3. Regarding the log transformation, how will researchers deal with the possibility that all the participants write the same number at the end (resulting in taking the logarithm of zero)?
4. In this case, the authors use decrease in log-variance as a proxy for "herding", but it seems that the process they are measuring here is more "convergence" rather than herding. To demonstrate herding, researchers would need to demonstrate some form of group polarisation, which cannot be captured with their measures here.

Minor:

1. Variables: While I applaud the use of equations to specify how measures are being calculated, it sometimes feels as though the authors are using them to distract readers from the actual simplicity of the measures (i.e., averaging, variance, etc.). To reach a wider audience, I would suggest that the authors use more plain language to explain the measures they are using rather than couch them in mathematical terms.

Author's Response to Decision Letter for (RSOS-191497.R0)

See Appendix B.

RSOS-191497.R1 (Revision)

Review form: Reviewer 1 (John Michael)

Do you have any ethical concerns with this paper?

No

Recommendation?

Accept in principle

Comments to the Author(s)

The authors have done a thorough job of engaging with the reviews and really improved the manuscript (as well as the study design itself). This will be a great contribution to the literature.

Review form: Reviewer 2

Do you have any ethical concerns with this paper?

No

Recommendation?

Reject

Comments to the Author(s)

It is still not clear, whether the paradigm and the study aim to address opinion dynamics or matters of truth (and uncertainty). This essential distinction has still not been grasped by the authors (e.g., p3 ll22.,p.4 ll.29 p.5ll.10) . In order to disambiguate the task, they should focus on truth-value free stimuli (in case, they want to observe opinion dynamics) or clearly specify which truth they are looking for and incentivizing accordingly (in case they want to observe wisdom of crowd effects). In either case, the stimuli at hand seem not suitable to separate the two different lines of interest: Right now, there are some prices based on a German auction sometime ago, but individuals are explicitly asked not to directly estimate prices achieved at the auction, but a generic market price (in Argentina) that has nothing to do with the outcomes of that auctions in Germany in the past: "Price will be estimated in USD which is the currency used to trade imported art in Argentina. Participants will be explicitly instructed to estimate the actual price, regardless of whether they liked each painting or not.". In the task as defined with an actual truth value (German auction price in the year xyz), this would be fine for matters of truth and participants should be incentivized accordingly. In case of opinion dynamics, the task should, however, still provide a reason why people should converge. Matters of confidence and expertise are about an assigned truth value, thus, this outcome can only make sense in a task with a truth value assigned. Variance total should not be the reason for selecting the task, because ambiguity may also increase the variance, but instead of variance in preferences and assigned value, this ambiguity will have variance of the actual truth.

The design is still problematic as variance in the ratings is still both dependent and independent variable in the most central hypothesis H4: It is not feasible by just splitting it in half to overcome the issue of apriori-relatedness of the two. If you define herding as the reduction in variance of the ratings from stage 1 to 3, then one would expect that there will be more chance for the high variance group to reduce from this high variance to a lower variance. While the authors realize

this to be a problem, their solution to artificially separate stimuli is in my eyes not sufficient to unravel this conflation of the two measures.

Likewise, the proposed between-design with individual comparators is unfortunately not overcoming this issue, as differences may as well be explained by not have stage 2.

Maybe, a different outcome of convergence/herding is warranted to ensure the dependent and independent variable are not conflated. This could be the speed of convergence or the likelihood of convergence. This would require both a totally different measurement and type of analyses.

My concerns with the log-transform shared by the other reviewers has not been alleviated and the authors chose to stick with it. Specifically, I wonder to what extent people will agree on the outcome after the group discussion, while an initial agreement is rather unlikely (as presented by the data of the pre-study), dependent on the incentives there might be a rationale why people may reinstate the group-agreed outcome and show zero variance at step 3. This is again indication that variance of the answers after discussion should not be the outcome of herding. As a measure of heterogeneity, one could as well use a different definition that would yield no reason for exclude zero variance cases (e.g. absolute difference to mean).

In sum, while some adjustments have been made to the introduction and the power-analysis, the proposed study design will not enable to reach the conclusions aimed for in the hypothesis. A conclusive design will need a different set-up unravelling the conflation between independent and dependent variable and clearly distinguishing between matters of truth and matters of opinion. This will yield a very different study rendering the pre-study not very informative and the current power analyses not helpful to determine the samples for a study that will help to address.

Review form: Reviewer 4

Do you have any ethical concerns with this paper?

No

Recommendation?

Accept in principle

Comments to the Author(s)

The reframing in the beginning has greatly helped with understanding the study and most concerns have been addressed. It would help, however, if the authors at some point were to post scripts of what they tell participants during the study (e.g., how are they exactly wording the directions at stage 2 to come to a consensus?).

Decision letter (RSOS-191497.R1)

Dear Dr Navajas,

The Editors assigned to your Stage 1 Registered Report ("Diversity of opinions and herding behaviour in uncertain crowds") have now received comments from reviewers. We would like you to revise your paper in accordance with the referee and editors suggestions which can be found below (not including confidential reports to the Editor). Please note this decision does not guarantee eventual acceptance.

Please submit a copy of your revised paper within three weeks (i.e. by the 22-Jul-2020). If we do not hear from you within this time then it will be assumed that the paper has been withdrawn. In exceptional circumstances, extensions may be possible if agreed with the Editorial Office in advance. We do not allow multiple rounds of revision so we urge you to make every effort to fully address all of the comments at this stage. If deemed necessary by the Editors, your manuscript will be sent back to one or more of the original reviewers for assessment. If the original reviewers are not available we may invite new reviewers.

When submitting your revised manuscript, you must respond to the comments made by the referees and upload a file "Response to Referees". Please use this to document how you have responded to the comments, and the adjustments you have made. In order to expedite the processing of the revised manuscript, please be as specific as possible in your response.

on behalf of Professor Chris Chambers (Registered Reports Editor, Royal Society Open Science)
openscience@royalsociety.org

Associate Editor Comments to Author (Professor Chris Chambers):

Associate Editor: 1

Comments to the Author:

The revised manuscript was returned to three of the four reviewers who assessed the original Stage 1 submission. Two reviewers (1 and 4) are broadly satisfied and recommend IPA (with Reviewer 4 requesting that the authors provide additional materials). Reviewer 2, however, continues to have major concerns about the study. Broadly speaking, the reviewer's primary objection is that the design is unable to answer the research question due to inappropriate choice of stimuli and conflation of key study variables. To me, these concerns seem reasonable and are well articulated. Based on my own reading of the manuscript, and the positive recommendations of the other two reviewers, I would however like to offer the authors a final opportunity to revise and address the comments of Reviewer 2. To avoid over-burdening the reviewers, a final accept/reject decision on a revised submission will be issued without returning the manuscript to

in-depth review. Please note that the revisions and/or rebuttal must be comprehensive and in-principle acceptance is not guaranteed.

Comments to Author:

Reviewer: 1

Comments to the Author(s)

The authors have done a thorough job of engaging with the reviews and really improved the manuscript (as well as the study design itself). This will be a great contribution to the literature.

Reviewer: 2

Comments to the Author(s)

It is still not clear, whether the paradigm and the study aim to address opinion dynamics or matters of truth (and uncertainty). This essential distinction has still not been grasped by the authors (e.g., p3 ll22.,p.4 ll.29 p.5ll.10) . In order to disambiguate the task, they should focus on truth-value free stimuli (in case, they want to observe opinion dynamics) or clearly specify which truth they are looking for and incentivizing accordingly (in case they want to observe wisdom of crowd effects). In either case, the stimuli at hand seem not suitable to separate the two different lines of interest: Right now, there are some prices based on a German auction sometime ago, but individuals are explicitly asked not to directly estimate prices achieved at the auction, but a generic market price (in Argentina) that has nothing to do with the outcomes of that auctions in Germany in the past: "Price will be estimated in USD which is the currency used to trade imported art in Argentina. Participants will be explicitly instructed to estimate the actual price, regardless of whether they liked each painting or not." In the task as defined with an actual truth value (German auction price in the year xyz), this would be fine for matters of truth and participants should be incentivized accordingly. In case of opinion dynamics, the task should, however, still provide a reason why people should converge. Matters of confidence and expertise are about an assigned truth value, thus, this outcome can only make sense in a task with a truth value assigned. Variance total should not be the reason for selecting the task, because ambiguity may also increase the variance, but instead of variance in preferences and assigned value, this ambiguity will have variance of the actual truth.

The design is still problematic as variance in the ratings is still both dependent and independent variable in the most central hypothesis H4: It is not feasible by just splitting it in half to overcome the issue of apriori-relatedness of the two. If you define herding as the reduction in variance of the ratings from stage 1 to 3, then one would expect that there will be more chance for the high variance group to reduce from this high variance to a lower variance. While the authors realize this to be a problem, their solution to artificially separate stimuli is in my eyes not sufficient to unravel this conflation of the two measures.

Likewise, the proposed between-design with individual comparators is unfortunately not overcoming this issue, as differences may as well be explained by not have stage 2.

Maybe, a different outcome of convergence/herding is warranted to ensure the dependent and independent variable are not conflated. This could be the speed of convergence or the likelihood of convergence. This would require both a totally different measurement and type of analyses.

My concerns with the log-transform shared by the other reviewers has not been alleviated and the authors chose to stick with it. Specifically, I wonder to what extent people will agree on the outcome after the group discussion, while an initial agreement is rather unlikely (as presented by the data of the pre-study), dependent on the incentives there might be a rationale why people may reinstate the group-agreed outcome and show zero variance at step 3. This is again indication that variance of the answers after discussion should not be the outcome of herding. As a measure of heterogeneity, one could as well use a different definition that would yield no reason for exclude zero variance cases (e.g. absolute difference to mean).

In sum, while some adjustments have been made to the introduction and the power-analysis, the proposed study design will not enable to reach the conclusions aimed for in the hypothesis. A conclusive design will need a different set-up unravelling the conflation between independent and dependent variable and clearly distinguishing between matters of truth and matters of opinion. This will yield a very different study rendering the pre-study not very informative and the current power analyses not helpful to determine the samples for a study that will help to address.

Reviewer: 4

Comments to the Author(s)

The reframing in the beginning has greatly helped with understanding the study and most concerns have been addressed. It would help, however, if the authors at some point were to post scripts of what they tell participants during the study (e.g., how are they exactly wording the directions at stage 2 to come to a consensus?).

Author's Response to Decision Letter for (RSOS-191497.R1)

See Appendix C

Decision letter (RSOS-191497.R2)

Dear Dr Navajas

On behalf of the Editor, I am pleased to inform you that your Manuscript RSOS-191497.R2 entitled "Diversity of opinions and herding behaviour in uncertain crowds" has been accepted in principle for publication in Royal Society Open Science.

You may now progress to Stage 2 and complete the study as approved. Before commencing data collection we ask that you:

- 1) Update the journal office as to the anticipated completion date of your study.
- 2) Register your approved protocol on the Open Science Framework (<https://osf.io/rr>) or other recognised repository, either publicly or privately under embargo until submission of the Stage 2 manuscript. Please note that a time-stamped, independent registration of the protocol is mandatory under journal policy, and manuscripts that do not conform to this requirement cannot be considered at Stage 2. The protocol should be registered unchanged from its current approved state, with the time-stamp preceding implementation of the approved study design. We recommend using the dedicated RR registration mechanism supported by the OSF: <https://osf.io/rr>

Following completion of your study, we invite you to resubmit your paper for peer review as a Stage 2 Registered Report. Please note that your manuscript can still be rejected for publication at Stage 2 if the Editors consider any of the following conditions to be met:

- The results were unable to test the authors' proposed hypotheses by failing to meet the approved outcome-neutral criteria.
- The authors altered the Introduction, rationale, or hypotheses, as approved in the Stage 1 submission.
- The authors failed to adhere closely to the registered experimental procedures. Please note that any deviations from the approved experimental procedures must be communicated to the editor immediately for approval, and prior to the completion of data collection. Failure to do so can result in revocation of in-principle acceptance and rejection at Stage 2 (see complete guidelines for further information).
- Any post-hoc (unregistered) analyses were either unjustified, insufficiently caveated, or overly dominant in shaping the authors' conclusions.
- The authors' conclusions were not justified given the data obtained.

We encourage you to read the complete guidelines for authors concerning Stage 2 submissions at <https://royalsocietypublishing.org/rsos/registered-reports#ReviewerGuideRegRep>. Please especially note the requirements for data sharing, reporting the URL of the independently registered protocol, and that withdrawing your manuscript will result in publication of a Withdrawn Registration.

Please note that Royal Society Open Science will introduce article processing charges for all new submissions received from 1 January 2018. Registered Reports submitted and accepted after this date will ONLY be subject to a charge if they subsequently progress to and are accepted as Stage 2 Registered Reports. If your manuscript is submitted and accepted for publication after 1 January 2018 (i.e. as a full Stage 2 Registered Report), you will be asked to pay the article processing charge, unless you request a waiver and this is approved by Royal Society Publishing. You can find out more about the charges at <https://royalsocietypublishing.org/rsos/charges>. Should you have any queries, please contact openscience@royalsociety.org.

Once again, thank you for submitting your manuscript to Royal Society Open Science and we look forward to receiving your Stage 2 submission. If you have any questions at all, please do not hesitate to get in touch. We look forward to hearing from you shortly with the anticipated submission date for your stage two manuscript.

on behalf of Professor Chris Chambers (Registered Reports Editor, Royal Society Open Science)
openscience@royalsociety.org

Author's Response to Decision Letter for (RSOS-191497.R2)

See Appendix D.

RSOS-191497.R3 (Revision)

Review form: Reviewer 1 (John Michael)

Is the manuscript scientifically sound in its present form?

Yes

Are the interpretations and conclusions justified by the results?

Yes

Is the language acceptable?

Yes

Do you have any ethical concerns with this paper?

No

Have you any concerns about statistical analyses in this paper?

No

Recommendation?

Accept with minor revision

Comments to the Author(s)

The authors have carried out the study and the analysis as planned, and the results are presented and discussed appropriately. A few minor issues might be worth addressing:

- The authors may want to indicate some of the study's limitations. For example, diversity of views may affect herding differently for different kinds of estimate. Here the participants were estimating the market value of paintings, and people's opinions about the market value of paintings do indeed influence the market value of those paintings. Things might be different for something that is independent of people's judgments, such as the weight of an object. Also, different kinds of diversity (e.g. in appearance, accent, profession, etc) which may affect herding in different ways.

- One concern: isn't it the case that where there is greater diversity (variability), there is also more room to herd? So assuming (hypothetically) that there are two groups of 5 people each, one highly variable in stage 1 and the other entirely homogenous. The people in the homogenous group have no possibility to herd at all, so of course the variable group will herd more (irrespective of which group values social information more). Or to change the scenario a bit: assume that the homogenous group is not entirely homogenous, but their estimates are clustered together more closely. Even if everyone in that group highly values social information and shifts their estimate, they won't shift it by much. I wonder whether you might be able to address this by checking in the data whether more people shift at all in more diverse groups (irrespective of how much they shift?

Decision letter (RSOS-191497.R3)

Dear Dr Navajas:

On behalf of the Editor, I am pleased to inform you that your Stage 2 Registered Report RSOS-191497.R3 entitled "Diversity of opinions promotes herding in uncertain crowds" has been deemed suitable for publication in Royal Society Open Science subject to minor revision in accordance with the referee suggestions. Please find the referees' comments at the end of this email.

The reviewers and Subject Editor have recommended publication, but also suggest some minor revisions to your manuscript. We invite you to respond to the comments and revise your manuscript. Below the referees' and Editors' comments (where applicable) we provide additional requirements. Final acceptance of your manuscript is dependent on these requirements being met. We provide guidance below to help you prepare your revision.

Please submit your revised manuscript and required files (see below) no later than 7 days from today's (ie 17-May-2022) date. Note: the ScholarOne system will 'lock' if submission of the revision is attempted 7 or more days after the deadline. If you do not think you will be able to meet this deadline please contact the editorial office immediately.

on behalf of Professor Chris Chambers (Associate Editor) and Chris Chambers
(Registered Reports Editor, Royal Society Open Science)
openscience@royalsociety.org

Associate Editor Comments to Author (Professor Chris Chambers):

Associate Editor: 1

Comments to the Author:

One of the original Stage 1 reviewers was available to evaluate the Stage 2 manuscript, and I have decided that we can proceed with an interim decision based on this review and my own reading. As you will see, the review is broadly positive and there should be few barriers to meeting the Stage 2 criteria. The reviewer offers some helpful suggestions concerning limitations to address in the Discussion, and a potential extra analysis to consider in order to tackle one such proposed limitation. In revising, please do not make any further changes to the Introduction and Methods.

Provided you are able to thoroughly address these points in a revision, full acceptance should be forthcoming without requiring further in-depth review.

Comments to Author:

Reviewer: 1

Comments to the Author(s)

The authors have carried out the study and the analysis as planned, and the results are presented and discussed appropriately. A few minor issues might be worth addressing:

- The authors may want to indicate some of the study's limitations. For example, diversity of views may affect herding differently for different kinds of estimate. Here the participants were estimating the market value of paintings, and people's opinions about the market value of paintings do indeed influence the market value of those paintings. Things might be different for something that is independent of people's judgments, such as the weight of an object. Also, different kinds of diversity (e.g. in appearance, accent, profession, etc) which may affect herding in different ways.

- One concern: isn't it the case that where there is greater diversity (variability), there is also more room to herd? So assuming (hypothetically) that there are two groups of 5 people each, one highly variable in stage 1 and the other entirely homogenous. The people in the homogenous group have no possibility to herd at all, so of course the variable group will herd more (irrespective of which group values social information more). Or to change the scenario a bit: assume that the homogenous group is not entirely homogenous, but their estimates are clustered together more closely. Even if everyone in that group highly values social information and shifts their estimate, they won't shift it by much. I wonder whether you might be able to address this by checking in the data whether more people shift at all in more diverse groups (irrespective of how much they shift?)

===PREPARING YOUR MANUSCRIPT===

one version should clearly identify all the changes that have been made (for instance, in coloured highlight, in bold text, or tracked changes);

If you have been asked to revise the written English in your submission as a condition of publication, you must do so, and you are expected to provide evidence that you have received language editing support. The journal would prefer that you use a professional language editing service and provide a certificate of editing, but a signed letter from a colleague who is a proficient user of English is acceptable. Note the journal has arranged a number of discounts for authors

using professional language editing services
(<https://royalsociety.org/journals/authors/benefits/language-editing/>).

===PREPARING YOUR REVISION IN SCHOLARONE===

-- If you are requesting an article processing charge waiver, you must select the relevant waiver option (if requesting a discretionary waiver, the form should have been uploaded, see 'File upload' above).

-- If you have uploaded any electronic supplementary (ESM) files, please ensure you follow the guidance at <https://royalsociety.org/journals/authors/author-guidelines/#supplementary-material> to include a suitable title and informative caption. An example of appropriate titling and

captioning may be found at https://figshare.com/articles/Table_S2_from_Is_there_a_trade-off_between_peak_performance_and_performance_breadth_across_temperatures_for_aerobic_sc_ope_in_teleost_fishes_/3843624.

Author's Response to Decision Letter for (RSOS-191497.R3)

See Appendix E.

Decision letter (RSOS-191497.R4)

Dear Dr Navajas:

I am pleased to inform you that your manuscript entitled "Diversity of opinions promotes herding in uncertain crowds" is now accepted for publication in Royal Society Open Science.

Please remember to make any data sets or code libraries 'live' prior to publication, and update any links as needed when you receive a proof to check - for instance, from a private 'for review' URL to a publicly accessible 'for publication' URL. It is also good practice to add data sets, code and other digital materials to your reference list.

Royal Society Open Science is a fully open access journal. A payment may be due before your article is published. Our partner Copyright Clearance Centre will contact the corresponding author about your open access options (if you have any queries regarding fees, please see <https://royalsocietypublishing.org/rsos/charges> or contact authorfees@royalsociety.org).

on behalf of Professor Professor Chris Chambers (Subject Editor).

Follow Royal Society Publishing on Twitter: @RSocPublishing
Follow Royal Society Publishing on Facebook:
<https://www.facebook.com/RoyalSocietyPublishing/>
Read Royal Society Publishing's blog:
<https://royalsociety.org/blog/blogsearchpage/?category=Publishing>

Appendix A

Review of RSOS-191497 “Diversity of opinions promotes herding in uncertain crowds”

This is an interesting study proposal, addressing how beliefs and confidences change as result of discussion. The topic is important and under-researched, and I would recommend going ahead. I do have a few concerns, which I hope could be addressed:

- Regarding the ‘logic, rationale and plausibility of hypotheses’ criterion, I have the following question. The main ‘counterintuitive’ hypothesis (H4) that the authors propose to test is that more diverse groups (that is, groups with more diverse initial estimates) will exhibit greater herding / convergence after discussion. The explanation for this hypothesis is that lack of diversity is a signal of strong common information component, hence individuals will stick with their own ‘distinctive’ information. I am not sure this argument goes through rigorously, and the authors should try to prove it using some model of how individuals update given their beliefs. More importantly, there seems to be an alternative explanation, namely, that high diversity groups are simply composed of more ignorant people (about art that is), who are therefore more willing to change opinions. How do the authors distinguish this explanation from their own?
- Regarding ‘methodology and statistics,’ I am wondering why log-variance of unlogged estimates is the appropriate measure of divergence of opinion instead of variance of logged estimates? The natural assumption with these price estimates is that their distribution is approximately log-normal. Why not estimate the variance of this distribution?
- Regarding ‘methodological detail,’ the protocol — especially in the discussion condition — needs to be fleshed out more. How will convergence to a common estimate be enforced? Why are the paintings shown for 40 seconds, rather than available during the entire discussion period?
- Terminology: The authors refer to ‘herding’ as the ‘uncoordinated convergence of thoughts or behaviors in a group.’ This to me sounds more like ‘groupthink.’ The term herding has acquired a more narrow theoretical interpretation (also called an ‘informational cascade’), as the process where person B copies person A, person C

then observes A and B, and defers to their majority judgment, ignoring the fact that B is just a copy of A, etc..

Appendix B

Reviewer: 1

Comments to the Author(s)

This is a very interesting and well-designed study.

The hypotheses are well-motivated and clearly stated, and the design is adequate to test them. The planned analyses seem appropriate and are stated with sufficient clarity and precision to limit 'undisclosed flexibility'. The sample size is justified. The methodology is also sufficient clear and detailed to enable replication.

Here are a few observations that may be useful to consider before finalizing the design:

We thank the reviewer for the positive feedback and the suggestions to improve the design. Below, we provide responses to each comment.

- One suggestion with respect to the idea that when people's opinions are close to each other they may assume they are relying on a common source of information, so they should have less reason to herd: Would you expect this to be influenced by what they know about the commonness or lack thereof of the source of info? In the current design of course they know that they all have seen the same paintings, but their beliefs about others' expertise may play a role. For example, if your estimate is wildly different from mine and I think you seem to have more expertise (e.g. because you have indicated this or because you appear confident), I may adjust to match you, but perhaps not if I think we have the same level of expertise. This is first of all just a general question and need not be taken into account here, but it could be a point for the discussion or for an experiment 2 (e.g. manipulating their beliefs about others' expertise or confidence). Or perhaps you could probe this by asking them how confident or competent the others in their group seemed.

The reviewer has made an interesting point about whether herding behaviour might depend on beliefs about other people's expertise. Based on this suggestion, we have now updated our Method and will ask participants to rate the perceived expertise of people in their group. Each individual will be assigned a numerical code (from 1 to 5), and at the end of the experiment, they will be asked "How much knowledge did other people in your group seemed to have about the art market?" Participants will be given 4 alternatives: "No Knowledge, Little Knowledge, A Moderate Amount of Knowledge, A Great Deal of Knowledge.". Every participant will answer this question about each of the 4 other individuals in the group. This will allow us testing the hypothesis that beliefs about expertise might correlate with herding behaviour, a secondary analysis that now appears in page 14.

Note, however, that we do not expect to have real experts. As the 'pilot' was done at the Tate Museum, people were at least self-selected in being interested in art, so we may not have had totally inexpert people. Overall, we do not plan to target experts therefore the role of actual expertise will not be substantial, which of course leaves open the question of projected or perceived expertise.

- In stage 2, what if they cannot come to a consensus?
- In stage 2, what if they write down different sums rather than the one collectively agreed?

Based on the reviewer's questions, we updated the description of the experimental procedure, so that these two questions are now addressed in the manuscript. Briefly, we now explicitly state that it will be identical to the one implemented in two of our previous experiments (Navajas et al., Nat. Hum. Behav., 2018; Navajas et al., Curr. Biol., 2019): we will ask participants to discuss their estimates in order to seek consensus, making clear that there are no advantages nor disadvantages (i.e., no bonus or penalty) associated with providing a collective estimate. If they were able to reach consensus, then all group members should write it on their own answer sheet. If they were not able to reach consensus, they should simply proceed with the experiment. However, we will discard groups that were not able to reach consensus on one or more paintings and proceed with data collection until we meet the target sample size. If participants wrote down different estimates rather than the one collectively agreed, we will consider that decision as if they had not reached consensus. This situation is extremely unusual and it did not happen in the Pilot Data nor in a similar previous experiment in a laboratory setting (Control Experiment in Navajas et al., 2018). This information now appears in pages 6-8 in sections "Procedure" and "Data Exclusion Criteria".

- In stage 3, 'participants will be invited to revise their initial estimates'. Does this prime them to revise/herd? (Even just using the word 'initial' may encourage revision.

We agree with the reviewer that the experimental procedure might invite people to revise estimates rather than insist. Indeed, this is part of the experimental design since we need groups to converge (H1) in order to study if they converge differently depending on their initial diversity (H4). In other words, while we agree with the reviewer that the design might naturally lead to herding behaviour, the test of our main hypothesis is not confounded by whether people revise estimates or insist on their opinion by instruction but studying whether groups herd differently depending on the initial diversity of opinions (something that, to our best knowledge, has not yet been empirically tested).

- Generally, I would worry a bit that they may not take stage 1 seriously, since they know they will have plenty of time to consider their response later and to take into account what they have heard from the others. It may be worth checking this for example by testing whether stage 1 responses in this design are less variable (with people just putting down a stock response each time) than in a treatment where there is only stage 1, or where they are told in stage 1 which paintings will be discussed with the group in stage 2. Or comparing to data from a previous experiment if you do not want to run this extra treatment. To be clear: I am not insisting that this is necessary; it is just a suggestion.

We thank the reviewer for the suggestion to control whether people take stage 1 seriously. Briefly, our experimental design posits that people will provide

price estimates for 8 paintings and then discuss only 4 of them. Because they do not know which paintings will be discussed and which ones will not, we expect that people will pay attention to stage 1 (as we perceived ourselves during the Pilot Experiment). However, and to test for a potential lack of attention to stage 1, we will compare the variability and accuracy of discussed and undiscussed items. This is now described in a secondary analysis.

To further explore this possibility, and following one of the reviewer's suggestions, we designed a Control Experiment where there will not be any social interaction between individuals. In this experiment, participants will not have the possibility to obtain more information through social interaction (or other means), so they should take stage 1 seriously. With these data, we will also compare the accuracy and variability of the stages 1 from the Control and Main experiments.

We also notice that, even if we found evidence that people did not pay full attention to stage 1, it will not represent a caveat to our results given that our main hypothesis is that diversity modulates herding (H4), and the lack of attention to stage 1 responses would apply to all groups, regardless of their initial diversity. We now describe the Control Experiment in page 8, and all Secondary Analyses in page 17.

- Will the stage 2 discussions be recorded? It could be interesting to see if there are some features of these interactions which predict herding and/or confidence boosts (The work by Fusaroli and Tylen and colleagues could provide some ideas here). It could also be useful to know if some of them take on the role of experts and influence the others in the group (perhaps because they really are more knowledgeable or perhaps just because they are more confident)

We agree with the reviewer that it would be interesting to record conversations and see if some features of the interactions correlate with herding behaviour. However, in this study we do not have any particular hypothesis about which specific features should correlate with herding. Given that these analyses would be entirely exploratory, we believe that they are better suited for a study with a different scope to the current one. For this reason, and for the sake of simplicity in data collection, we will not record the conversations.

Reviewer: 2

Comments to the Author(s)

The manuscript “Diversity of opinions promotes herding in uncertain crowds” describes a quasi-experimental design to test whether initial high variance in opinions will lead to more private conformity after group decision. While I was rather enthused by the description of the approach in the abstract and the question how variance in opinion may alter social influence, reading through the manuscript vastly dampened my enthusiasm: The introduction lacks the necessary rigor and ignores most classic literature and the presented design is unconvincing and unclear.

We thank the reviewer for the thoughtful feedback. Based on the referee’s comments, we have now reframed the introduction and cited the relevant literature so that it better represents the scope of our study. We have also modified several aspects of our experimental design in order to address the reviewer’s concerns. These include the design of a Control Experiment, the addition of a new hypothesis (H6), and the re-calculation of sample size. Below, we provide a point-by-point reply to each comment.

Introduction:

While I wonder why the authors cite Milgrim’s obedience studies, as they only marginal relate to the research question. Instead, they should have considered the classic distinction between informational and normative influence (Deutsch & Gerard, 1955). This is crucial as it will introduce the differentiation between private and public conformity and makes clear that the presented study will be rather scrutinizing informational influence, where people infer knowledge from other sources while being unaware of any objective standard (Sherif, 1935), instead of normative influence, where they will maintain their private opinion on a clear, objective truth while being pressured to publicly conform with the crowd (Asch, 1956).

The authors are completely ignoring this rather large part of the literature that deals with ambiguous stimuli (what the authors name uncertainty) and informational influence.

Based on the referee’s comment, we have reframed the introduction to highlight the classic conceptual distinction between informational and normative herding. We also explain better why, in line with recent developments in the field, the two types of herding are not seen as exclusive, and that herding can be driven by both types.

Here we are in line for instance with Baddeley, M. (2010). Herding, social influence and economic decision-making: socio-psychological and neuroscientific analyses. *Philosophical Transactions of the Royal Society B: Biological Sciences*, 365(1538), 281-290.

“The most powerful explanations for herding and social influence emphasize the dual roles played by reason and emotion. Herding and imitation in economic and financial decision-making may reflect a social learning process

but this will be moderated by emotions and by socio- psychological traits determining receptivity to social influence.”

Following the reviewer’s comment, we now describe more concisely how our study fits with the existing literature.

The introduction is now structured as follows:

*** In the first paragraph, we define the concept of herding as “the uncoordinated convergence of thoughts or behaviours” (Raafat, Chater & Frith, 2009; Baddeley, 2010) and cite classic studies demonstrating the existence of this effect (e.g., Sherif, 1935; Asch, 1951; Deutsch & Gerard, 1955; Moscovici et al., 1969). As requested by the reviewer, we removed the reference to Milgram (1963).**

*** In the second paragraph, we explain that, in those studies, participants had no uncertainty about the majority opinion, a situation which is rarely the case outside the laboratory. We argue that whether, and to what extent, humans infer and react to the variance of group opinions is still unknown.**

*** Then, in the third paragraph, we argue that our study aims to fill that empirical gap. In that paragraph, we now introduce the classic distinction between informational and normative herding, citing the paper by Deutsch & Gerard, and explicitly stating that our study may well reflect an informational herding effect, though it can be modulated by normative herding (reference now given to review by Baddeley, 2010)**

*** The fourth and fifth paragraphs describe the rationale underlying the two opposite hypotheses regarding how variance might modulate herding behaviour. In the sixth paragraph, we briefly describe and justify the selection of our task.**

All these changes now appear in pages 3-5.

Furthermore, it appears to me that arguments for the diversity hypothesis (H4) constitute a post-hoc rationalization of trends in the pilot data. While, provided there is an exact true answer unknown to the participants, people should be cautious not to rely on low-reliability shared cues, convergence of estimates mostly indicates correct inference and not overreliance on wrong cues. This is also reflected in the author’s argumentation in the following paragraph that similarity should also boost confidence. To determine an overreliance on correlated cues, one would need to observe the convergence of cues, not of estimates. A good discussion on how certainty and diverging answers can be used as cues for correct answers, please see Moussaid , Kämmer, Analytis, Neth (2013). Speaking even more profoundly against the author’s rationale, in the case presented, there is no objective truth and thus no cue to follow to exploit wisdom of crowds: The monetary value of art is a very philosophical question and results of an auction in Germany, that is some while ago (and the time is not specified in the manuscript), is not a very good indicator of the current market price for these paintings in Argentina. Moreover, the authors argue that they deliberately chose a scenario, in which there is an uncertainty of truth.

Alternatively, if they want to study wisdom of crowds, they should ask for estimates on subjects, for which there is an objective truth clearly unknown to participants (length of (not very famous) rivers, heights, etc.).

Regarding the reviewer's concern about a potential post-hoc rationalization of a trend in the pilot data, we now discarded this possibility by proposing two-tailed testing in H4 (as suggested by the reviewer in a comment below). In other words, we are testing not only for an effect in the direction of the observed trend in the pilot data, but also for the opposite effect. Moreover, we acknowledge in the introduction that existing theories deliver opposite predictions and explain the rationale for both of them. This resulted in an increase in the target sample size which we have included in the revised proposed method.

Regarding the suggestion to look at cues instead of estimates, we agree with the reviewer that, in order to infer over-reliance on shared cues, one could potentially use the strategy developed in Moussaid et al. (2013). However, unlike the experiment proposed here, the paper by Moussaid and colleagues used a task where there was large heterogeneity in expertise. In fact, one necessary condition to infer cue reliability from confidence is that subjects should display a high correlation between confidence and accuracy (see Figure 1B in Moussaid et al.). In contrast, our task –where participants try to estimate the market price of paintings– was designed so that participants were similarly uninformed and uncertain as one another. Indeed, the pilot data suggests that there will be no correlation between confidence and accuracy ($r=.05$, $p=.17$). Given that confidence will not signal accuracy, and that participants will be similarly uninformed, the only way to infer reliability is by looking at the distance between estimates (i.e. the 'majority effect' in Moussaid et al., 2013), which is exactly what we measure here by computing the variance of opinions.

Finally, and regarding the reviewer's comments about our rationale, we now clarified several aspects of our experiment:

1) We now explicitly state that the proposed experiment asks about the actual price of paintings as it appears in the German auction website, namely, we ask for an unquestionably objective number. In that sense, the price of art is not a philosophical but an economic question (the value of art is a philosophical question). We will update the prices at the time of the experiment and reinforce the instruction that subjects should estimate those prices independently of whether or not they like the paintings. This avoids framing the task as a discussion of personal willingness to pay or tastes.

2) We now make explicit that this experiment is not about the 'wisdom of crowds', as we do not study whether the aggregation of individual estimates is close to the truth. Instead, we study whether groups with different variance herd differently.

3) The reviewer asks whether we could have used a more classical task where people estimate quantities based on general knowledge such as length of rivers or heights of landmarks. We now provide a reason why our task based on estimating prices is better suited to address H4. Briefly, in order to study how herding correlates with variance, we need to generate an experimental setting where there is heterogeneity in group variance, and the pilot data suggests that we will indeed observe a wide range of groups log-variances (mean \pm S.D. = 9.63 ± 1.96). To compare these values with a more classical task, we re-analysed the data of a previous experiment performed by our group (Navajas et al., Nat. Hum. Behav., 2018) and found that the variance of opinions elicited by that task was substantially lower (mean \pm S.D. = 3.58 ± 1.41 , rank sum test, $z=14.2$, $p=8 \times 10^{-46}$). We also observed that the heterogeneity of group variances was significantly smaller (Fisher test for equal variances, $F(839,75)=.53$, $p=1.7 \times 10^{-5}$). In other words, this analysis suggests that using a more classical task would go against our main aim of studying how group variance modulates herding.

We now added these clarifications and the re-analysis of Navajas et al. (2018) in the Method section (page 18).

Design:

It is not described, how participants are expected to reach consensus and how/ by whom this info will be put forward in the discussion phase. Techniques of reaching consensus might be quite different (averaging the individual estimates/ 'expert' decision/ voting) and the results might be quite variable dependent on , for instance, order of presentation of single estimates. As such, 40s appears to be not a lot of time for discussion and reaching consensus.

Following the reviewer's comment, we now modified and described with more detail the procedure associated to the instructions of reaching consensus. The procedure will be identical to the one implemented in two of our previous experiments (Navajas et al., Nat. Hum. Behav., 2018; Navajas et al., Curr. Biol., 2019): we will ask participants to discuss their estimates in order to seek consensus, making clear that there are no advantages nor disadvantages associated with providing a collective estimate. If they were able to reach consensus, then all group members should write it on their own answer sheet. We will not instruct participants to implement any specific technique of reaching consensus (e.g., averaging), but let them freely discuss a collective estimate.

Based on the referee's intuition that 40s might not be enough to induce consensus, we increased the time to 2 minutes. This is twice as much as what we gave in a previous experiment involving numerical estimates (Navajas et al., 2018) and the same of amount of time we gave in an experiment involving reaching consensus about complex and controversial moral issues (Navajas et al., 2019). Based on the success of those two previous experiments, we expect that 2 minutes will be sufficient to induce consensus in the current study.

We now expanded the Method section to further describe the instructions.

Furthermore, the incentivization of the participants is not described. Will they receive just a show-up fee or will they be incentivized to estimate as close as possible to the German auction?

We thank the reviewer for noticing that the economic incentives were not clearly described. We now explicitly state that participants will be incentivised with a show-up fee of 100 ARS, which is the standard rate of short experiments at Universidad Torcuato Di Tella (less than 30 minutes long).

The reason why we do not incentivise accuracy is because previous research suggests that it might distort herding behaviour. For example, one study has shown that the extent to which individuals converge to each other in social interactions depend on whether incentives are individual or collective (Bazazi et al., PLOS ONE, 2019). In order to avoid this additional level of complexity in our setup, which leads to research questions that are orthogonal to the main scope of this study, we will not incentivise accuracy.

Our experience with similar previous experiments (involving the estimation of numerical estimates with social interactions in a laboratory setting) tells us that this incentive procedure will be sufficient to produce high-quality individual, collective, and revised ratings (see Control Experiment in Navajas et al., 2018).

The within-control is not fully convincing to me: I am not sure that regression to the mean effects are fully controlled for, when you take just half of the items as variance indicator: the group's characteristics will remain the same (as you could show with the high correlations). Instead, a between-group control with a non-collective task will be a much more credible control for the group effects.

Based on the reviewer's suggestion to have a between-group control with a non-collective task, we now added a Control Experiment to our proposal. This experiment will be similar in structure to the Main Experiment but will not involve any social interaction between participants. Individuals will be presented with the paintings, then perform an unrelated task (i.e., answering a series of general-knowledge questions), and finally provide revised ratings. With this setup, we will then create 'virtual quintets' by randomly grouping individuals. These data can then be used to compute the variables 'Herding' using Equation [4] and 'Diversity' using Equation [6] and estimate the correlation between those variables in a context with no social influence. Critically, as the reviewer suggests, all effects of regression to the mean should also be present in the Control Experiment. We will then evaluate an additional hypothesis (H6 in the current manuscript) which is that the correlation between diversity and herding in the Main Experiment will be greater than the one observed in the Control Experiment. We thank the reviewer for this suggestion which we believe will strengthen the conclusions of our study.

The Control Experiment is now described in page 8, and H6 now appears in page 14.

I am also not sure about the log-transformation performed – this will cause the exclusion of 0 variance groups and other statistical models may account for issues of bounded values of the outcome variable.

The reviewer is indeed correct that the log-transformation proposed in this study leads to the exclusion of zero-variance groups. However, we also notice that this situation is extremely unusual: we had less than 1% of zero-variance quintets in an experiment with N=280 groups and 8 general-knowledge questions (Navajas et al., 2018), and zero cases in the pilot experiment consisting in N=19 groups and 8 paintings. However, if we happened to observe a group that produces individual estimates with zero variance, we will discard all data proceeding from that group and continue with data collection until we meet the target sample size. We now clarify this procedure in the Method section in page 18.

While I appreciate that pre-test data was used and that the presented power analysis considers the group as UE, the authors should not assume one-tailed testing, as the introduction made clear that opposite hypothesis is also discussed in literature.

Based on the reviewer suggestion, we now re-computed the power analysis using two-tailed testing. This led to an increase in the target sample size to N=650 with 95% power. We thank the reviewer for this suggestion and describe this procedure in pages 17-18.

Reviewer: 3

Comments to the Author(s)

This is an interesting study proposal, addressing how beliefs and confidences change as result of discussion. The topic is important and under-researched, and I would recommend going ahead. I do have a few concerns, which I hope could be addressed:

We thank the reviewer for the positive feedback and the constructive comments. Below, we address the four concerns raised by the referee.

1) Regarding the 'logic, rationale and plausibility of hypotheses' criterion, I have the following question. The main 'counterintuitive' hypothesis (H4) that the authors propose to test is that more diverse groups (that is, groups with more diverse initial estimates) will exhibit greater herding / convergence after discussion. The explanation for this hypothesis is that lack of diversity is a signal of strong common information component, hence individuals will stick with their own 'distinctive' information. I am not sure this argument goes through rigorously, and the authors should try to prove it using some model of how individuals update given their beliefs. More importantly, there seems to be an alternative explanation, namely, that high diversity groups are simply composed of more ignorant people (about art that is), who are therefore more willing to change opinions. How do the authors distinguish this explanation from their own?

Based on the feedback received during this revision, we modified the framing so that we now test two alternative hypotheses: H4a (less diverse groups will exhibit greater herding) and H4b (more diverse groups will exhibit greater herding). This reflects our acknowledgement to conflicting and opposite predictions made by different theories. On the one hand, theoretical research in social psychology predicts that social influence should decrease as the distance between social sources become larger (e.g., Social Comparison Theory). On the other hand, qualitative formulations in political science (e.g., De Tocqueville, 1835) and quantitative theories in other fields have shown conditions where more variable estimates signal the existence of unshared information which can be efficiently pooled by groups (e.g., Kao et al., 2014). In those situations, individuals have strong reasons to herd given that social information is not redundant with private information (e.g., Bohren et al., 2009). While we accept that these papers presented models in conditions which are arguably different to the task developed here, we believe that constructing a quantitative theoretical formulation of optimal belief updating is beyond the scope of the current study, which aims at empirically testing how variance modulates herding behaviour.

The reviewer also raised an interesting point about whether and how we can distinguish an explanation of why diverse groups could herd more based on differences in expertise. Briefly, individuals in more diverse groups could be more prone to herd because they are formed by people who have lesser expertise in art. However, as stated in the 'Participants' subheading in the 'Method' section, we will only recruit participants who have no academic training in arts and zero experience in the art business. Participants with any

detected expertise in arts will be discarded if recruited incidentally. As such, our proposed design aims to test the hypothesis making sure that variations in expertise are kept to minimal.

Of course, these leaves open the question of projected or perceived expertise. Although all participants will have similar (lack of) expertise about art prices, they could still think that some people have more expertise than others. To test for this possibility, we will ask participants to rate the perceived expertise of people in their group. Each individual will be assigned a numerical code (from 1 to 5), and at the end of the experiment, they will be asked “How much knowledge did other people in your group seemed to have about the art market?” Participants will be given 4 alternatives: “No Knowledge, Little Knowledge, A Moderate Amount of Knowledge, A Great Deal of Knowledge.”. Every participant will answer this question about each of the 4 other individuals in the group. This will allow us testing the hypothesis that beliefs about expertise might correlate with herding behaviour, a secondary analysis that now appears in page 7.

In terms of the relationship between diversity and expertise, we note that more diverse groups will naturally produce estimates with more individual error. Indeed, we observed a significant correlation between diversity and mean absolute error in the Pilot Data ($r=.53$, $p=5\times 10^{-7}$). However, if those differences in error were indeed based on differences in expertise, one should also expect a negative correlation between diversity and confidence, and a negative correlation between confidence and absolute error. However, we observed that confidence does not negatively scale with diversity ($r=.15$, $p=.20$) nor with actual error ($r=.17$, $p=.15$), suggesting that differences in error could not be attributed to differences in expertise.

2) Regarding 'methodology and statistics,' I am wondering why log-variance of unlogged estimates is the appropriate measure of divergence of opinion instead of variance of logged estimates? The natural assumption with these price estimates is that their distribution is approximately log-normal. Why not estimate the variance of this distribution?

The reviewer asks an interesting question: what is the rationale underlying the computation of log-variances as opposed to taking the variance of log-estimates? The reason underlying this procedure is that the variance of log-estimates is a positive quantity (i.e. it cannot take negative values), and since our hypothesis involves measuring a reduction in variance (i.e., the convergence of opinions), then it is subject to 'floor effects': the reduction in variance cannot exceed the initial variance. To prevent our analysis to suffer from 'floor effects' we estimate the reduction in group diversity using log-variances, which are unbounded. This way, by taking the difference in log-variances, we are measuring the *relative change* in group diversity, thus making a fair comparison between groups with different degree of initial variance. We explain this procedure in the Method section (description of H1).

3) Regarding 'methodological detail,' the protocol — especially in the discussion condition— needs to be fleshed out more. How will convergence to a common

estimate be enforced? Why are the paintings shown for 40 seconds, rather than available during the entire discussion period?

Based on the reviewer's questions, we now provide more details about the experimental design. The procedure will be identical to the one implemented in two of our previous experiments (Navajas et al., Nat. Hum. Behav., 2018; Navajas et al., Curr. Biol., 2019): we will ask participants to discuss their opinions in order to seek consensus, making clear that there are no advantages nor disadvantages associated with providing a collective estimate. We will not instruct participants to implement any specific technique of reaching consensus (e.g., averaging), but let them freely discuss a collective estimate. If they were able to reach consensus in two minutes, then all group members should write it on their own answer sheets. In case they did not reach consensus, participants will simply indicate so in their answer sheets and proceed with the experiment. In other words, participants will not be enforced to reach consensus and will not have any economic incentives to do so. Based on our previous experience performing similar experiments in a laboratory setting (e.g., 'Control Experiment' in Navajas et al., 2018), we expect that 40 seconds should be enough time to induce consensus about numerical estimates. However, and to make sure that participants will have the opportunity to freely discuss their estimates, we extended that time to two minutes. All this information now appears in the Method section (subheading 'Procedure').

Following the reviewer's comment, and although discussions will last a maximum of 2 minutes, we will display the paintings to the participants for the duration of the entire discussion. Thanks for this suggestion.

4) Terminology: The authors refer to 'herding' as the 'uncoordinated convergence of thoughts or behaviors in a group.' This to me sounds more like 'groupthink.' The term herding has acquired a more narrow theoretical interpretation (also called an 'informational cascade'), as the process where person B copies person A, person C then observes A and B, and defers to their majority judgment, ignoring the fact that B is just a copy of A, etc.

We agree with the reviewer that several authors, most of them proceeding from economics and finance, have used the word 'herding' in the narrow context of sequential learning from social signals. However, we also notice that the term has been extensively used with other meanings in fields as dissimilar as social psychology, epidemiology, political science, and genetics (for a comprehensive review, see Table 1 in Rafaat, Chater & Frith, "Herding in humans", TiCS, 2009). For this reason, and to avoid misunderstandings about the scope of our study, we now explicitly define herding in page 11 (i.e., in the description of H1) clarifying that we use the word "herding" as in the "convergence of beliefs in a group through local interactions", a definition which is consistent with previous research (e.g., Trotter, 1908; Rafaat et al., 2009; Baddeley, 2010; Stallen et al., 2012).

Reviewer: 4

Comments to the Author(s)

The authors want to test a hypothesis of "herding". They suggest that in groups with greater variance, greater decrease of variance across individual valuations will occur when other members' valuations are known, but in groups with lower variance, knowing other group members' valuations will increase confidence.

We thank the reviewer for the comments, feedback, and suggestions to improve the manuscript. Below, we provide a point-by-point reply to all the concerns raised by the referee.

Major concerns:

1. The researchers explicitly tell groups that they must converge to a value in their discussion. This naturally will lead to a decrease in variance. This seems to me to be a major design flaw that explains away all effects the researchers might find.

Based on the reviewer's comment, we now clarified two main aspects of our experimental design.

First, we better explained the instructions provided to the participants in the second stage of the experiment. Briefly, we will ask them to discuss their estimates with others and try reaching consensus. We will explicitly state that there will be no advantages nor disadvantages associated to reaching consensus, and they will not have any economic incentives to do so. In case they did not reach consensus, participants will simply indicate so in their answer sheets and proceed with the experiment. In other words, participants will not be obliged to provide collective estimates in the second part of the experiment.

Second, as the reviewer correctly suggests, asking participants to attempt reaching consensus will probably lead to a decrease in variance, something that we indeed expect to observe. This process of convergence, which was previously described in the literature, will be our starting point (i.e., H1) to test our main hypothesis (H4). More specifically, this paper is not simply about showing that groups might converge in group settings (something that has been already shown elsewhere) but studying whether groups herd differently depending on the initial diversity of opinions (something that, to our best knowledge, has not yet been empirically tested). In other words, observing support for H1 speaks in favor of our experimental design: we need groups to converge in order to study if they converge differently depending on their initial diversity.

Based on the reviewer's comment, we now clarify that H1 is a necessary condition to test our main hypothesis (H4).

2. Given the natural inclination of people to shift their valuations towards the values of others (e.g., Zaki, Schirmer, & Mitchell, 2011), I am not sure why one shouldn't expect that people's values converge when they can share their valuations in a group. Moreover, one might expect that when estimates within a group are close together (i.e., exhibiting low variance), there is naturally less of an inclination to shift one's values because one is close enough anyway. This will naturally lead to lower levels of convergence in groups with lower levels of variance at t_0 ; this is something that cannot be rectified by simply looking at ratings on a log scale, nor, if we assume stability in intragroup variance across all painting, can it be rectified by calculating diversity of the non-discussed paintings.

Here we wish to clarify two points, where there may be part-agreement, part misunderstanding. First, our hypothesis is not that “one should expect that people’s values converge when estimates are close”. We do agree with the reviewer and with the extensive literature in social psychology in expecting convergence. The main point of this study is that we expect a *different degree of convergence with higher diversity* (see H4 for details).

While the reviewer may find this result natural, the literature offers conflicting predictions on whether diversity should increase or decrease herding behavior. We now explicitly state this design with “alternative hypotheses” in the description of H4.

In our first submission and here, we also agree with the reviewer that, in pursuing this hypothesis, there is a threat of circularity that needs to be avoided. This is exactly the caution we raised in the first submission and which we have addressed in the design of the experiments and proposed analyses. As reviewers 1 to 3 have also recognized, the design and analyses are a sound way to address (H4), and with the additional clarifications and additions now added, we hope that this will have now lifted the misunderstanding.

3. Regarding the log transformation, how will researchers deal with the possibility that all the participants write the same number at the end (resulting in taking the logarithm of zero)?

Based on the reviewer’s comment, we now explain how we would deal with zero-variance groups. We first notice that this situation is extremely unusual: we had less than 1% zero-variance quintets in an experiment with $N=280$ groups and 8 general-knowledge questions (Navajas et al., Nat. Hum. Behav., 2018), and zero cases in the pilot experiment consisting in $N=19$ groups and 8 paintings.

However, in case we observed a group where all five individuals provided the same estimate, thus leading to zero variance, we will exclude that group from

the analysis and proceed with data collection until we meet the target sample size. We now clarify this procedure in the Method section (“Data Exclusion Criteria”, pages 8-9).

4. In this case, the authors use decrease in log-variance as a proxy for "herding", but it seems that the process they are measuring here is more "convergence" rather than herding. To demonstrate herding, researchers would need to demonstrate some form of group polarisation, which cannot be captured with their measures here.

We agree with the reviewer that the process we are measuring here is precisely the convergence of beliefs in group settings. However, we also notice that this is exactly what previous research have previously defined as “herding”. For example, an extensive review of different theoretical frameworks has defined “herding” as “the alignment of thoughts or behaviours of individuals in a group through local interactions” (Rafaat, Chater, & Frith, “Herding in humans”, TiCS, 2009). While we recognize that the term “herding” has been used with a different connotation in other disciplines (e.g., in economics, as the imitation process observed in sequential learning paradigms), we believe that our task captures a process of alignment that fits with the definition given, for example, in Rafaat et al. (2009).

Finally, and regarding the group polarisation phenomenon, we argue that becoming (on average) more extreme is a different process to the convergence of opinions. In other words, the group polarisation bias looks at changes in the average group ratings while the process of convergence we study here looks at changes in group variance. While it would be certainly interesting to study how social influence modulates changes in the average price of paintings, we believe that it addresses a different question to the one posed in our manuscript and would be beyond the scope of the current study.

Based on the reviewer’s comment, we now explicitly state our working definition of “herding” as the “convergence of beliefs in a group” (page 11, see the description of H1).

Minor:

1. Variables: While I applaud the use of equations to specify how measures are being calculated, it sometimes feels as though the authors are using them to distract readers from the actual simplicity of the measures (i.e., averaging, variance, etc.). To reach a wider audience, I would suggest that the authors use more plain language to explain the measures they are using rather than couch them in mathematical terms.

Following the reviewer’s suggestion, we now made sure that all hypotheses in the manuscript are written in plain language, using no equations (see titles of H1 to H6). However, we would prefer keeping the mathematical definitions of the relevant variables for the sake of avoiding undisclosed flexibility in the analysis (given that this is a registered report).

This way readers can choose whether they would like to know the precise definition of variables such as “diversity” or “herding” while, at the same time, they could also avoid them.

Appendix C

Dear Associate Editor and Reviewers,

Thank you for your handling and reviewing our paper entitled "Diversity of opinions and herding behaviour in uncertain crowds" (Manuscript ID RSOS-191497.R2) which has been granted in-principle-acceptance on 27 July 2020.

Following the protocol registration at the Open Science Framework (<https://osf.io/eqnyx/>) and the definition of the deadline for submission of the Stage 2 manuscript (26 July 2021), we have now been forced to revise and resubmit our protocol. This is because the experiment proposed in our study involved face-to-face conversations which go against current COVID-19 regulations.

Our previous task involved face-to-face conversations in groups of 5 individuals. This condition is still unpermitted by current protocols at the proposed data collection site (Universidad Torcuato Di Tella, Argentina) as well as in the proposed back-up site (LMU, Germany). Given the current worldwide circumstances, the speed of vaccination, and the rate of new confirmed cases in Argentina and Germany, we are now certain that we won't be able to launch that behavioural study by the proposed deadline. For example, Universidad Torcuato Di Tella in Argentina has already confirmed that research facilities will remain closed until at least August 2021.

For these reasons, we hereby propose to modify the protocol in our study so that data collection could happen in virtual rooms using Zoom (<https://zoom.us/>). In the revised manuscript, you will find that the only differences with the original protocol appear in pages 6-8, where we describe the Experimental Procedure. (All relevant changes have been written in red.) Briefly, we will now ask participants to join a virtual meeting and conversations will occur in groups of five using the "Breakout Rooms" tool in Zoom. The nature of the task, script of the instructions to participants, data exclusion criteria, hypotheses, proposed analyses, control experiment, and power analyses remain unchanged from the accepted manuscript.

We regret this unfortunate situation and having to request this modification to the accepted protocol. We are also very grateful for your time and the feedback provided throughout the review process and look forward to your evaluation of the modified protocol.

With kindest regards,

On behalf of all authors,

Joaquin Navajas

Appendix D

Dear Prof. Chambers,

Thank you for accepting our stage 1 protocol entitled “*Diversity of opinions and herding behaviour in uncertain crowds*” and for the invitation to submit a stage 2 Registered Report at *Royal Society Open Science*. Please find enclosed our manuscript.

Briefly, we have been able to closely follow the methodological procedure preregistered in the stage 1 report and have collected data across two studies: a main study (N=650) consisting in 130 group deliberations and a control study (N=100) where there was no social influence.

Below, we detail the main points related to this submission:

- We have made all raw and processed data available, as well as all codes, through the Open Science Framework (<https://osf.io/n5zd6/>). The reference to this link appears in page 19 under the section “Data and codes availability”.
- The reference to the approved stage-1 pre-registered protocol (<https://osf.io/s89w4>) appears in the abstract (page 2) and in the first sentence of the Results (page 20).
- We confirm that none of the data submitted here has been collected before IPA and that we performed all main and secondary analyses that were clearly specified in the pre-registered protocol.
- We also confirm that the completed experiments have been executed and analyzed in the manner originally approved with any unforeseen changes in those approved methods and analyses clearly noted.
- We added two new sections (Results and Discussion) with two new figures (Figures 3 and 4) where we report and discuss the findings produced by following our protocol.
- We added a section in the Results where we separately report three secondary analyses that were not preregistered in the original protocol. This is under a section called “Non-preregistered analyses” in page 23.
- The content in the Introduction and Methods sections were left unchanged, and the only modifications were related to changes from future tense (e.g., “We will collect data from...”) to past tense (e.g., “We have collected data from...”).

We also need to highlight two modifications in the present submission:

- There has been one change in author order: Prof. Ophelia Deroy is now the 5th and last author while before she was 4th in the list. This change has been approved by all coauthors of the paper.
- We also would like to kindly request a change in the title of the paper to “*Diversity of opinions promote herding in uncertain crowds*”. We believe this less ambiguous title will be more appropriate given the results that we report in this submission.

We appreciate your time and look forward to your response.

Thank you for receiving our manuscript,

On behalf of all authors,

Joaquin Navajas

Universidad Torcuato Di Tella

Buenos Aires, Argentina

Appendix E

Dear Prof. Chambers,

Thank you for reviewing our stage 2 manuscript entitled “*Diversity of opinions promotes herding in uncertain crowds*” and for the invitation to submit a revised version for consideration at *Royal Society Open Science*. Please find enclosed our manuscript.

Briefly, we have now addressed the two points raised by the reviewer:

- We have added a paragraph in the Discussion where we elaborate on the limitations of the present study in terms of the task selection and definition of “diversity”. This paragraph is highlighted in red in page 26.
- As suggested by the reviewer, we have now performed a third non-preregistered analysis where we studied whether the effect of diversity on herding persists if herding was defined as a categorical variable. A logistic regression of diversity on herding also provided evidence for the hypothesized effect. The paragraph where we present this finding is highlighted in red in page 24.
- We have updated our codes in the Open Science Framework to add this new analysis.
- As requested by the editorial guidelines, we have not modified any other section of the paper.

We appreciate your time and look forward to your response.

Thank you for receiving our manuscript,

On behalf of all authors,

Joaquin Navajas

Universidad Torcuato Di Tella
Buenos Aires, Argentina